# A fungal pathogen manipulates phytocytokine signaling for plant infection

Chenlei Hua [1] ✉, Lisha Zhang [1], Annick Stintzi [2], Andreas Schaller [2], Hui-Shan Guo [3] ✉ & Thorsten Nürnberger [1] ✉

Phytocytokines, hormone-like plant peptides, play crucial roles in immune regulation and development. Phytosulfokine (PSK), known to mediate plant growth, also modulates plant pattern-triggered immunity (PTI). Here, we demonstrate that VdSCP8, a small cysteine-containing effector from *Verticillium dahliae*, functions as a virulence-promoting protein that suppresses PTI in *Arabidopsis thaliana* and *Nicotiana benthamiana*. Apoplastic SCP8 suppresses immune activation mediated by leucine-rich repeat ectodomain pattern recognition receptors. SCP8 virulence and immunosuppressive activities require PHYTOSULFOKINE RECEPTOR 1 (PSKR1), which binds PSK and forms a complex with co-receptor BAK1 to suppress PTI. Our findings underscore SCP8's role in PTI suppression, facilitating PSKR1-BAK1 complex formation and promoting PSK accumulation through a plant subtilase. These results highlight how a multi-host plant pathogen manipulates PTI through enhancing immunosuppressive PSK signaling.

In nature, the interaction between plants and microbes plays a pivotal role in shaping ecosystem dynamics and stability. While many microbial species form mutually beneficial symbioses with plants, phytopathogenic microbes exploit plants as hosts to complete their life cycles, causing significant damage to agricultural crops and natural plant habitats. To successfully colonize plant hosts, these microbes employ large sets of effectors to divert plant nutrients for their own reproduction and to evade or suppress host immune responses[1].

To counter microbial attacks, plants have evolved pattern-recognition receptors (PRR) located on the cell surface that activate immunity. PRRs detect conserved microbial patterns, triggering pattern-triggered immunity (PTI)[2]. Upon ligand binding, PRRs recruit membrane-localized co-receptors like somatic embryogenesis receptor kinases (SERKs), and transduce extracellular signals to downstream signaling components including receptor-like cytoplasmic kinases[3], mitogen-activated protein kinases[4] and transcription factors, culminating eventually in the expression of defense genes, the production of antimicrobial compounds and priming for enhanced disease resistance[5].

While plant pathogenic microbes use effectors to manipulate host immune signaling pathways—from PRR activation to defense gene expression - plants possess intracellular immune receptors that detect potentially harmful effector activities upon their delivery into host plant cells. This effector-triggered immunity (ETI) in concert with early PTI activation is assumed to establish robust immunity against microbial infections[6–8].

In uninfected plants, PTI activation is under negative control by at least two phytocytokines[9–14]. Rapid alkalinization factor (RALF), recognized by the malectin domain-containing receptor kinase FERONIA[15], and phytosulfokine (PSK), a tyrosine-sulfated pentapeptide recognized by the leucine-rich repeat (LRR) domain-containing receptor kinase PSKR1[11,12,16]. Several fungal plant pathogens produce and secrete RALF-like peptides to suppress PTI[17], illustrating how manipulation of host defense negative regulatory mechanisms aids in successful host infection.

*Verticillium dahliae* (*Vd*) is a soil-borne fungus with a wide host range[18], including model plants like *Nicotiana benthamiana* and *Arabidopsis thaliana* (hereafter Arabidopsis)[19,20]. In Arabidopsis, *Vd*-

[1]Department of Plant Biochemistry, Centre for Plant Molecular Biology (ZMBP), Eberhard-Karls-University of Tübingen, Tübingen, Germany. [2]Department of Plant Physiology and Biochemistry, University of Hohenheim, Stuttgart, Germany. [3]State Key Laboratory of Microbial Diversity and Innovative Utilization, Institute of Microbiology, Chinese Academy of Sciences, Beijing, China. ✉e-mail: chenlei.hua@zmbp.uni-tuebingen.de; guohs@im.ac.cn; nuernberger@uni-tuebingen.de

derived patterns chitin, NECROSIS AND ETHYLENE-INDUCING PEP-TIDE 1-LIKE PROTEIN (NLP)-derived peptide nlp20, and POLYGALACTURONASE-derived peptide fragment pg9 activate PTI[21–25] which engage PRR-complexes such as LYSIN-MOTIF RECEPTOR KINASE 5 (LYK5)/CHITIN ELICITOR RECEPTOR KINASE 1 (CERK1), RECEPTOR-LIKE PROTEIN 23 (RLP23)/SUPPRESSOR OF BRASSINOS-TEROID INSENSITIVE 1 (BRI1)-ASSOCIATED KINASE (BAK1)-INTER-ACTING RECEPTOR KINASE 1 (SOBIR1), and RLP42/SOBIR1, respectively. Both RLP23/SOBIR1 and RLP42/SOBIR1 depend on SERK co-receptors to activate downstream signaling[21,25], akin to receptor-like kinases like FLAGELLIN-SENSITIVE 2 (FLS2) or EF-TU RECEPTOR (EFR)[26,27]. In contrast, several *Vd* effectors have been identified that suppress chitin-induced plant immunity[28–30]. Our previous research identified several small cysteine-containing protein effectors (VdSCP, hereafter SCP) from *Vd*, which either suppress or activate plant immunity-associated responses[31,32]. While some SCPs localize to the nucleus to target Arabidopsis transcription factors, effectors SCP8, SCP9, and SCP10 were found to accumulate extracellularly at the plant-pathogen interface[33]. SCP8 is regulated by a *Vd* enolase for host infection[34], but its mode of action *in planta* remains unclear. Here, we demonstrate that SCP8 functions as a fungal virulence factor that promotes host plant infection by manipulating PSK signaling, ultimately leading to suppression of PTI.

## Results

### PSK signaling activation is required for *Verticillium dahliae* (*Vd*) infection

To study fungal infection-associated changes in the host transcriptome, we conducted RNA sequencing (RNAseq) analyses of *Nicotiana benthamiana* plants infected with various *Vd* strains over multiple time points. Our results revealed significant upregulation of genes encoding *N. benthamiana* SOBIR1, PSK, and its receptors PSKR1 and PSKR2 during *Vd* infection (Fig. 1a and Supplementary Table 1). However, these genes did not exhibit notable upregulation when *N. benthamiana* was treated with microbe-associated molecular patterns (MAMPs) such as flg22[35], nlp20[36], and chitin, or with water as control (Fig. 1b). Based on these findings, we hypothesized that unknown *Vd* effectors might interfere with PSK signaling to manipulate PTI, thereby promoting infection and colonization. To test this, we infected several Arabidopsis genotypes, including Col-0, the PSKR1 mutant *pskr1-3*[24], the PSKR1 and PSKR2 double mutant *pskr1-3pskr2-1*[37], and the SERK3 (also known as BAK1) and SERK4 (also known as BKK1) double mutant *bak1-5bkk1-1*[26] with the *Vd* strain. The results demonstrated that *bak1-5/bkk1-1* plants were more susceptible to *Vd* infection, while both *pskr1-3* and *pskr1-3pskr2-1* genotypes exhibited enhanced resistance to *Vd* infection compared to the wild type Col-0 (Fig. 1c, d). Consistently, PSK gene overexpression in Arabidopsis enhanced susceptibility to *Vd*

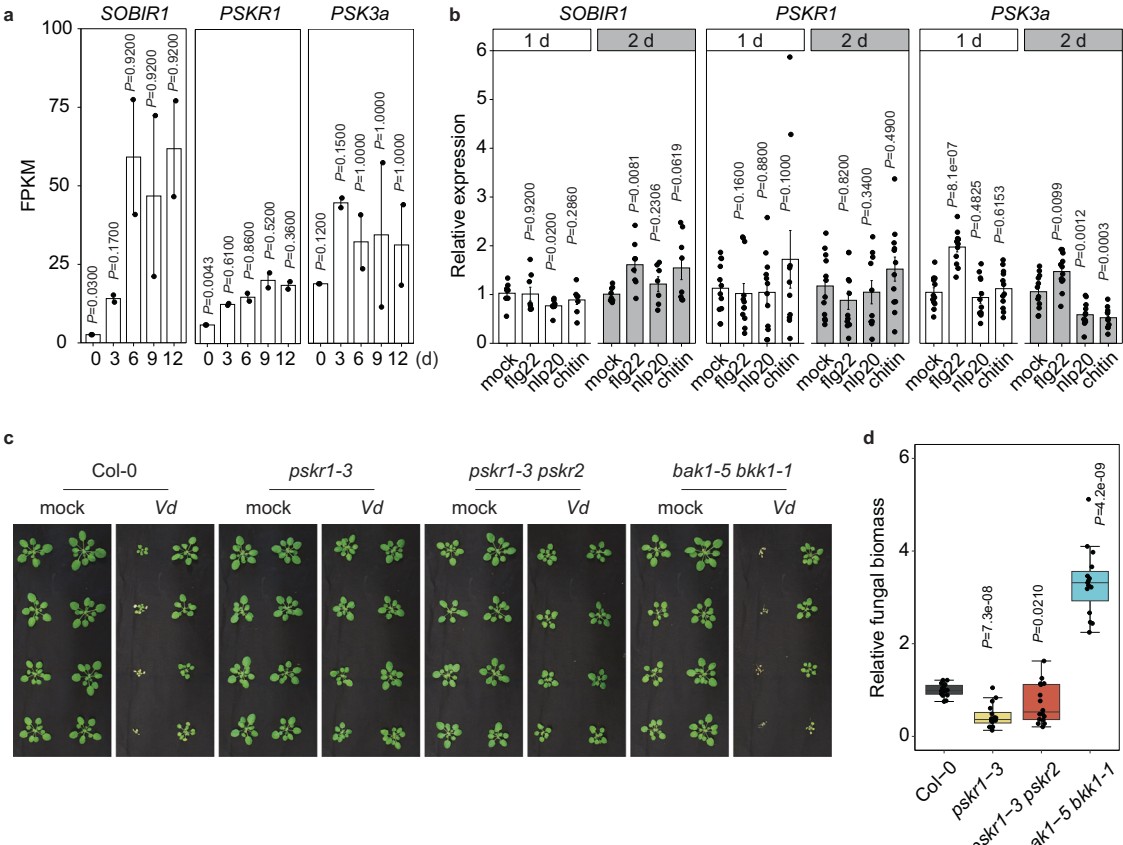

**Fig. 1 | PSK signaling activation is required for *Vd* infection. a** Gene expression in *Nicotiana benthamiana* following infection with *Verticillium dahliae* was assessed by RNA sequencing (RNA-seq) at 0, 3, 6, 9, and 12 days post-inoculation (dpi). Expression levels were quantified as fragments per kilobase of transcript per million mapped reads (FPKM). **b** Gene expression in *N. benthamiana* following pattern treatment at 1 day and 2 days post infiltration (1 μM concentration). Water served as mock control. **c** Representative images of Arabidopsis Col-0, *pskr1-3*, *pskr1-3pskr2-1*, and *bak1-5bkk1-1* lines infected with *Vd* at 20 days post inoculation. Water treatment served as mock controls. **d** Quantification of *Vd* biomass in infected Arabidopsis plants from panel (**c**). DNA was extracted from whole plants above the soil, and the relative fungal biomass was assessed by comparing Ct values of *Vd actin* and *Arabidopsis EF1α* genes. For **a** and **b**, bars represent means ± SE of two (**a**) and three (**b**, n = 8 for *SOBIR1*, n = 12 for *PSKR1* and *PSK3a*) independent experiments. Data points are indicated as black dots. Statistical differences to any time points (**a**) or to mock treatments at the respective time (**b**) are indicated (two-sided Student's *t*test). **d** Data points are indicated as black dots from four independent experiments (n = 16) and plotted as box plots (center line, median; bounds of box, the first and the third quartiles; whiskers, 1.5 x IQR; error bar, minima and maxima). Statistical differences between Col-0 and the indicated mutants were analysed using a two-sided Student's *t*test.

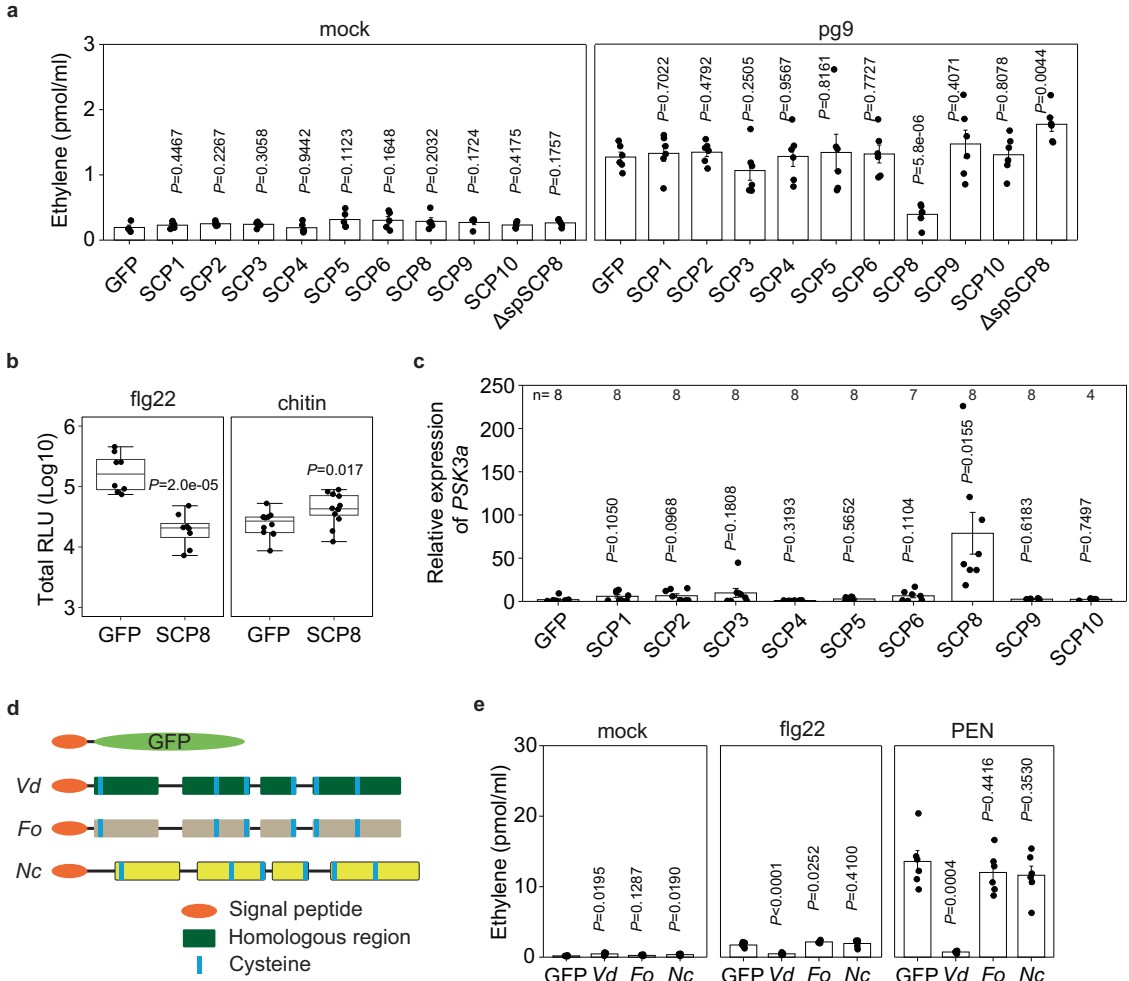

**Fig. 2 | SCP8 acts as a suppressor of BAK1 dependent immune responses and an inducer of *PSK* gene expression. a** Ethylene production in *RLP42*-transgenic *N. benthamiana* plants infiltrated with agrobacteria encoding SCP genes, followed by treatment with pg9 (1 μM). SCP proteins were expressed as N-terminal fusions with a leader peptide derived from Arabidopsis PR1 to facilitate protein secretion. SCP8 without N-terminal PR1 leader peptide (ΔspSCP8) was also expressed in this assay. Agroinfiltration was performed 3 days before the pg9 treatment, and ethylene levels were measured 2 h after pg9 treatment. **b** Reactive oxygen species (ROS) burst in *N. benthamiana* leaves infiltrated with agrobacteria encoding *SCP8* genes, followed by treatment with flg22 (1 μM) and chitin (1 μM). **c** *PSK3a* gene expression in *N. benthamiana* leaves infiltrated with agrobacteria encoding *SCP* genes. Leaves were collected at 3 days post agroinfiltration for RNA isolation and quantitative PCR (qPCR) analysis. **d** Schematic diagram illustrating SCP8 and its homologs from

different fungi used for agro-infiltration in *N. benthamiana*. *Vd* represents *Verticillium dahliae* SCP8 (*VdSCP8*), *Fo* denotes *Fusarium oxysporum* SCP8 (*FoSCP8*), and *Nc* indicates *Neurospora crassa* SCP8 (*NcSCP8*). **e** Ethylene production in *N. benthamiana* leaves infiltrated with agrobacteria expressing SCP8 homologous genes following treatment with flg22 (2 μM) and PEN (200 μg/ml). Ethylene levels were measured 2 h post elicitor treatments. For **a**, **c**, and **e**, bars represent means ± SE of two or three independent experiments (**a**, n = 5 for mock, *n* = 6 for pg9; **e**, n = 6). Data points are indicated as black dots. **b** Data points are indicated as black dots from three independent experiments (*n* = 8 for flg22, *n* = 12 for chitin) and plotted as box plots (center line, median; bounds of box, the first and the third quartiles; whiskers, 1.5 x IQR; error bar, minima and maxima). Statistical differences to the response observed in GFP-expressing plant are indicated (two-sided Student's *t*test).

infection (Supplementary Fig. 1). These findings underscore the importance of PSK signaling in successful plant infection.

## VdSCP8 suppresses BAK1-dependent immune responses and induces *PSK* gene expression

*Vd* SCP effectors have been shown to modulate plant defenses[31,32]. To investigate the potential immune-modulatory activities of these effectors and their sites of action, we generated N-terminal fusions of the Arabidopsis *PR1* signal peptide with SCP-coding sequences for *35S*-promoter-driven transient overexpression in Arabidopsis RLP42-transgenic *N. benthamiana* plants. *At*RLP42 is a receptor-like protein (RLP) PRR that features a leucine-rich repeat ectodomain for recognizing a peptide fragment (pg9) derived from fungal polygalacturonases[24], and it lacks a cytoplasmic kinase domain. Of the nine *Vd* SCPs tested, only SCP8, a 194-amino acid protein with six cysteine residues[32], strongly reduced ethylene production in pg9-

treated leaves (Fig. 2a). However, intracellular transient over-expression of SCP8 (ΔspSCP8) did not reduce pg9-induced ethylene production (Fig. 2a), suggesting that apoplastic localization of SCP8 is essential for its immunosuppressive activity. Additionally, SCP8 was found to suppress bacterial flagellin peptide flg22-induced ROS burst, which is mediated by the LRR receptor kinase FLS2[38]. Interestingly, SCP8 overexpression did not affect the fungal chitin-induced ROS burst, which is mediated by the binding of chitin to the lysin-motif ectodomains of receptor kinases LYK5/CERK1 (Fig. 2b). These findings suggest that SCP8 specifically suppresses plant immunity triggered by LRR-type PRRs, which require co-receptor BAK1 for signal transduction[5].

Furthermore, we observed that *N. benthamiana* leaves expressing SCP8 exhibited a significant upregulation of the *PSK3a* gene (Supplementary Table 2) compared to GFP-expressing controls and other transgenic plants expressing SCP effectors (Fig. 2c). This observation

supports the hypothesis that SCP8 activates PSK signaling to suppress plant immunity, particularly that mediated by LRR-type PRRs.

*SCP8*-related genes are widespread across Ascomycota, including both pathogenic fungi like *Fusarium oxysporum* and *Magnaporthe oryzae*, as well as non-pathogenic fungi like *Neurospora crassa*. Sequence alignments of the deduced amino acid sequences from pathogenic and non-pathogenic fungi revealed six conserved cysteines and four domain blocks (Fig. 2d, Supplementary Table 2). SCP8 is distantly related to the *Vd* elicitor PevD1[39] and to the *Alternaria alternata* major allergen Alt a1[40].

To test whether the PTI-suppressive functions are conserved among SCP8-like proteins from other fungi, we transiently expressed *F. oxysporum FoSCP8* and *N. crassa NcSCP8*-encoding sequences in *N. benthamiana* leaves, and assessed the ethylene production induced by pattern recognition. While GFP-expressing control leaves produced significant ethylene within 2 h after treatment with flg22 or the *Penicillium*-derived elicitor (PEN)[41], leaves expressing *Vd*SCP8 did not respond to either elicitor. However, leaves expressing *FoSCP8* and *NcSCP8* (Fig. 2e) showed no reduction in ethylene production, suggesting that SCP8 exhibits species-specific effector activity.

## SCP8 is a *Vd* virulence factor

To evaluate the contribution of *SCP8* to fungal virulence, *SCP8* knock-out mutants were generated using homologous recombination[42]. Two independent gene-knockout mutants, *Δscp8-1* and *Δscp8-2*, which showed no obvious defects in fungal development (Supplementary Fig. 2), were selected for host infection assays. Additionally, we introduced a *SCP8-GFP* gene fusion, driven by either the native or the constitutive OliC promoter[43], into the *Δscp8-1* mutant to generate genetically complemented strains C-Pnative and C-Polic, respectively.

Infection assays on *Vd* host plants (cotton, Arabidopsis, and *N. benthamiana)* showed that cotton infected with *Vd* wild-type strain V592 displayed severe wilt symptoms by 20 dpi, whereas cotton plants inoculated with *Δscp8-1-* or *Δscp8-2* strains exhibited significantly reduced disease progression (Fig. 3a, b). Similar outcomes were observed for Arabidopsis and *N. benthamiana* (Fig. 3c, d, and e). Independent complementation with both native and constitutive promoter-driven *SCP8* gene constructs restored virulence in *Δscp8-1* on cotton hosts (Fig. 3a, b).

Furthermore, we generated Arabidopsis transgenic lines stably over-expressing Arabidopsis *PR1 signal peptide-SCP8-GFP* constructs (Supplementary Fig. 3a and 3b). Two lines, T-6 and T-12 with similar SCP8 expression levels, exhibited increased susceptibility to *Vd* infection compared to wild type Col-0 (Fig. 3f). Altogether, these findings demonstrate that SCP8 is required for full virulence of *Vd*.

Next, we treated leaves of Col-0 and transgenic lines T-6 and T-12 with flg22, nlp20, pg9 and chitin, followed by measurements of immune responses including ethylene production, ROS burst and MAPK activation. As shown in Fig. 4a, ethylene production induced by nlp20 and pg9 was lower in the T-6 and T-12 transgenic lines compared to wild-type Col-0 at 3 hpi, while responses to flg22 and chitin showed no significant differences. Since flg22 and chitin typically induce relatively low ethylene production in Arabidopsis compared to nlp20 and pg9, we optimized the treatment method by infiltrating flg22 and chitin and extending the time of treatment to 4 h. Notably, the optimized flg22 treatment significantly increased ethylene production in Col-0, reaching levels higher than those observed in T-6 and T-12, whereas chitin treatment did not show the same effect (Supplementary Fig. 3c, d). Similarly, the ROS burst in transgenic plants was significantly reduced compared to wild-type Col-0 in response to flg22, nlp20, or pg9 but not to chitin (Fig. 4b). Additionally, MAPK activation by flg22, nlp20, and pg9 was significantly attenuated in SCP8 transgenic lines compared to wild type, while chitin-induced MAPK phosphorylation showed no difference between Col-0 and SCP8 transgenic lines (Fig. 4c).

## SCP8 facilitates PSK precursor digestion in cooperation with plant subtilases

To investigate how SCP8 suppresses plant defense, we performed a mass spectrometry (MS)-based analysis to identify SCP8-interacting proteins in *N. benthamiana*. Proteins were isolated from *N. benthamiana* leaves expressing SCP8-GFP or free GFP by immunoprecipitation using anti-GFP agarose beads, followed by MS analysis. The analysis revealed that the majority of peptides detected specifically in the SCP8-GFP sample corresponded to the subtilase (SBT) Niben101Scf00726g02006.1, a protein closely related to tomato SBT3[44] and Arabidopsis SBT1 and SBT3 families[45,46] (Supplementary Table 3). Subtilases are known to play a role in processing propeptides that lead to the formation of mature phytocytokines, such as systemin and PSK[47,48]. The interaction between SCP8 and *N. benthamiana* SBT was further confirmed by co-immunoprecipitation (Co-IP) assays in *N. benthamiana* leaves co-expressing GFP-tagged SCP8 and myc-tagged SBT1 (Fig. 5a) and Bimolecular Fluorescence Complementation (BiFC) assays (Supplementary Fig. 4a) as described previously[7]. SCP8 also interacted with other plant subtilases, including Arabidopsis SBT1.3 (AtSBT1.3) and SBT1.9 (AtSBT1.9), as well as cotton SBT1 (GhSBT1) (Supplementary Fig. 4b).

To evaluate whether SBT activity contributes to SCP8 function, we co-expressed two SBT inhibitors, SPI-1 from Arabidopsis[49] and EPI1 from the oomycete pathogen *Phytophthora infestans*[50] along with SCP8 (or free GFP as a control) in *N. benthamiana*. The ROS burst induced by various doses of flg22 was subsequently measured. As shown in Fig. 5b, co-expression of SCP8 with either SPI-1 or EPI1 significantly impaired SCP8-mediated suppression of plant defense, indicating that SBT activity is required for SCP8's role in modulating plant immunity.

We hypothesized that SCP8 might enhance or activate subtilase activity, leading to more phytocytokine production (such as PSK) and, consequently, suppression of PTI. To test this, we examined PSK precursor processing in the presence of SCP8 and subtilase inhibitors in *N. benthamiana*. The *PSK3a* gene, which is up-regulated by SCP8 (Fig. 2c), was synthesized as a $His_6$-GFP fusion construct and transiently expressed in *N. benthamiana* (Supplementary Fig. 5a). Western blot analysis using a GFP antibody revealed two bands between 25 and 35 kDa, consistent with the cleavage of the $His_6$-GFP-tagged PSK3a precursor (PSK3a-HGFP). Based on the predicted protein sizes (Supplementary Fig. 5a), the larger band likely represents the full-length precursor, while the smaller, weaker band likely corresponds to the cleaved PSK3a-HGFP. Co-expression of PSK3a-HGFP with the subtilase inhibitor SPI-1 led to accumulation of the full-length precursor (Supplementary Fig. 5b), suggesting that endogenous plant subtilase activity was inhibited by SPI-1. A slight increase in full-length PSK3a-HGFP was also observed with co-expression of EPI1 (Supplementary Fig. 5b). Similar results were obtained using the Arabidopsis PSK1 precursor (Supplementary Fig. 5b). Notably, SCP8 expression led to a marked reduction in full-length PSK3a-HGFP compared to the precursor expressed alone or with SPI-1, demonstrating that SCP8 promotes PSK precursor cleavage *in planta* (Fig. 5c). Surprisingly, SCP8-mediated cleavage of the PSK precursor was not suppressed by either SPI-1 or EPI1 (Fig. 5c and Supplementary Fig. 5c). The observed loss of immune suppression with co-expression of SCP8 and subtilase inhibitors may be due to the inhibition of other subtilases that do not process PSK precursors, given that SCP8 interacts with multiple plant subtilases (Supplementary Fig. 4b).

To determine whether SCP8 directly cleaves the PSK3a precursor or acts through plant subtilases, we purified PSK3a-HGFP, C-terminally $His_6$-tagged SBT1 from *N. benthamiana*, and C-terminally myc- and $His_6$-tagged SCP8 from *Pichia pastoris* (Supplementary Fig. 6). As shown in Fig. 5d, incubation of PSK3a-HGFP with either SBT1 or SCP8 protein at room temperature for 15 h did not result in significant cleavage compared to the PSK3a-HGFP control, although some self-

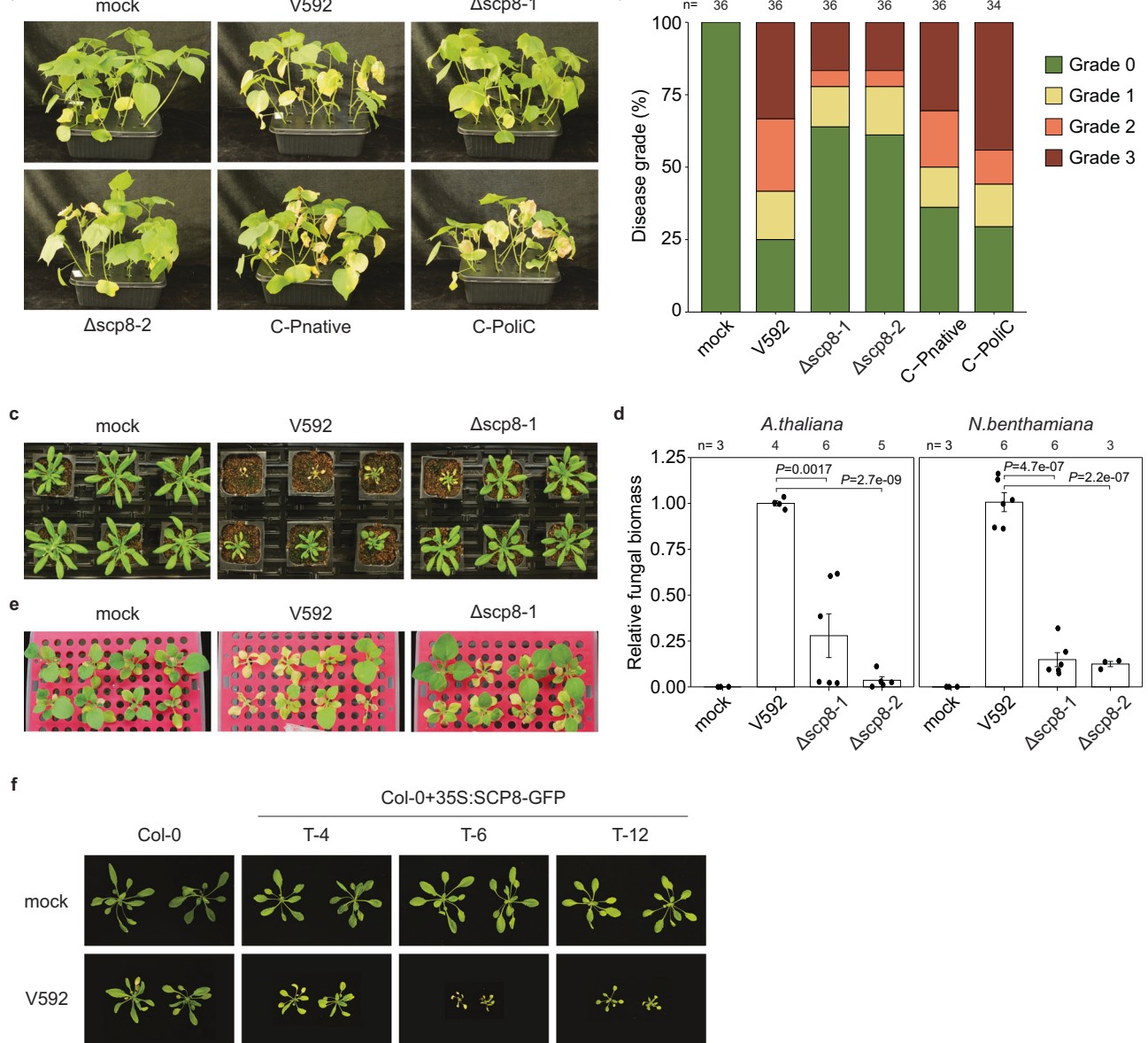

**Fig. 3 | Plants infected with *Vd* wild-type strain V592, *SCP8* gene knock-out strains *Δscp8-1*, *Δscp8-2*, and gene complementation transformants C-Pnative and C-Polic. a** Images of cotton seedlings infected with *Vd* strains (V592, *Δscp8-1*, *Δscp8-2*, C-Pnative, and C-Polic) were captured at 20 days post inoculation. **b** Disease quantification in cotton plants is presented. **c** Arabidopsis plants infected with *Vd* strains were photographed at 20 days post inoculation. **d** *Vd* biomass quantification in infected Arabidopsis and *N. benthamiana* is shown. Bars represent means ± SE of two independent experiments, each consisting of two or three biological replicates. Individual data points are shown black dots. Statistical differences to *Vd* wild-type strain V592 are indicated (two-sided Student's *t*test). **e** Images of *N. benthamiana* infected with *Vd* strains were taken at 20 days post inoculation. **f** Transgenic Arabidopsis plants expressing SCP8, infected with *Vd*, were photographed at 20 days post inoculation.

degradation was observed. However, when PSK3a-HGFP was incubated with both SBT1 and SCP8, clear digestion occurred under the same conditions. Reducing the incubation time to 3 h yielded similar results (Supplementary Fig. 7), supporting the hypothesis that SCP8 enhances PSK generation by interacting with plant subtilases.

### SCP8 mediated immune suppression requires activation of PSK-PSKR1 signaling

Consistent with the role of PSK in SCP8-mediated immune suppression, flg22-induced ethylene production in Col-0 was strongly suppressed when PSK was co-infiltrated, while chitin-induced ethylene production was unaffected. In contrast, ethylene production was restored in the PSK receptor mutant, *pskr1-3* (Fig. 6a). To further confirm the involvement of PSK signaling in SCP8 activity, we

attempted to express *SCP8* in Arabidopsis PSK-receptor mutants, including *pskr1-3*, *pskr1-3/pskr2-1*, and *pskr1/psy1r*. However, we were unable to obtain stable transgenic lines, suggesting that continuous SCP8 expression disrupts essential plant functions.

As an alternative approach, we harvested apoplastic fluids from *N. benthamiana* plants expressing SCP8-GFP or free GFP and infiltrated it, along with flg22 or pg9, into Col-0 and *pskr1-3* leaves. In Col-0 leaves, co-infiltration of apoplastic fluid containing SCP8 with either flg22 or pg9 resulted in a more than 50% reduction in ethylene production compared to co-infiltration with apoplastic fluid containing GFP. In *pskr1-3* leaves, however, this suppression was reduced, with difference-of-differences (DoD) values of 0.23 ($P = 0.0649$) for flg22 and 0.32 ($P = 0.3336$) for pg9, indicating statistically insignificant attenuation of PTI suppression (Fig. 6b). Since apoplastic fluids may contain other

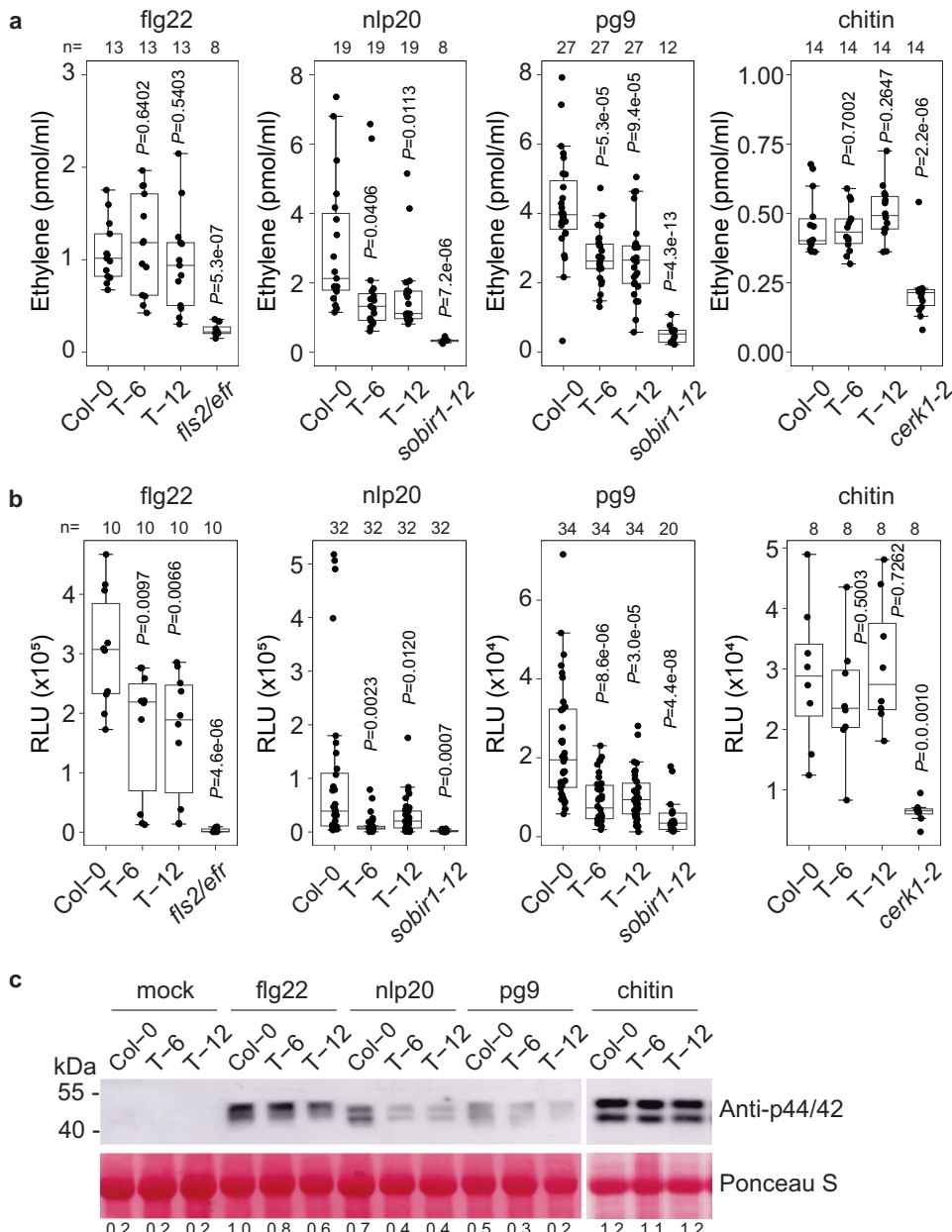

**Fig. 4 | Immune responses of transgenic *Arabidopsis* expressing SCP8 upon pattern treatments. a** Ethylene production. **b** ROS burst. For **a** and **b**, data points are indicated as black dots from two-four independent experiments and plotted as box plots (center line, median; bounds of box, the first and the third quartiles; whiskers, 1.5 x IQR; error bar, minima and maxima). Statistical differences between Col-0 and the indicated mutants were analysed using a two-sided Student's *t*test.

**c** MAPK activation in *Arabidopsis* Col-0, T-6, and T-12 plants following treatment with flg22, nlp20, pg9 and chitin (all patterns applied at 1 μM concentrations). Protein levels were approximated by measuring band intensities from western blots using ImageJ software. Band intensities were normalized to those of the corresponding loading controls. The protein level in flg22-treated Col-0 samples was set to 1.0 for normalization.

components that could affect ethylene levels, we repeated the experiment using purified recombinant His-tagged SCP8 protein produced in *P. pastoris* (Supplementary Fig. 6c). When co-infiltrated with flg22 or pg9 into Col-0 leaves, purified SCP8 reduced ethylene production compared to GFP controls. In contrast, SCP8 had no suppressive effect in *pskr1-3*, with DoD values of 0.36 ($P = 0.0421$) for flg22 and 0.71 ($P = 0.0253$) for pg9 (Fig. 6c), indicating a significant genotype-dependent difference. These findings suggest that SCP8 relies on PSK signaling to dampen host immune responses.

PSK is known to activate the formation of PSKR1-BAK1 complexes[16], which suppress PTI[11,12,51]. To test whether SCP8 induces the formation of PSKR1-BAK1 complexes, we infiltrated recombinant SCP8 into Arabidopsis plants stably expressing *p35S::AtPSKR1-GFP*[52]

and performed immunoprecipitation using GFP-trap agarose beads. Co-Immunoprecipitation of *At*BAK1 with *At*PSKR1-GFP was detected at 5–30 min after SCP8 treatment, but not following water or nlp20 treatments alone (negative controls) (Fig. 6d). As expected, PSK treatment resulted in high levels of precipitated *At*BAK1 (positive controls). Similar results were obtained in *N. benthamiana* transiently co-expressing *p35S::AtPSKR1-GFP* and *p35S::AtBAK1-myc* constructs, confirming that SCP8 also promotes the formation of *At*PSKR1-*At*BAK1 complexes in a heterologous system (Supplementary Fig. 8).

In summary, we conclude that SCP8 promotes the digestion of PSK precursors by interacting with plant subtilases, such as SBT1, ultimately facilitating immune suppression through PSKR1-BAK1 complex formation (Fig. 6e).

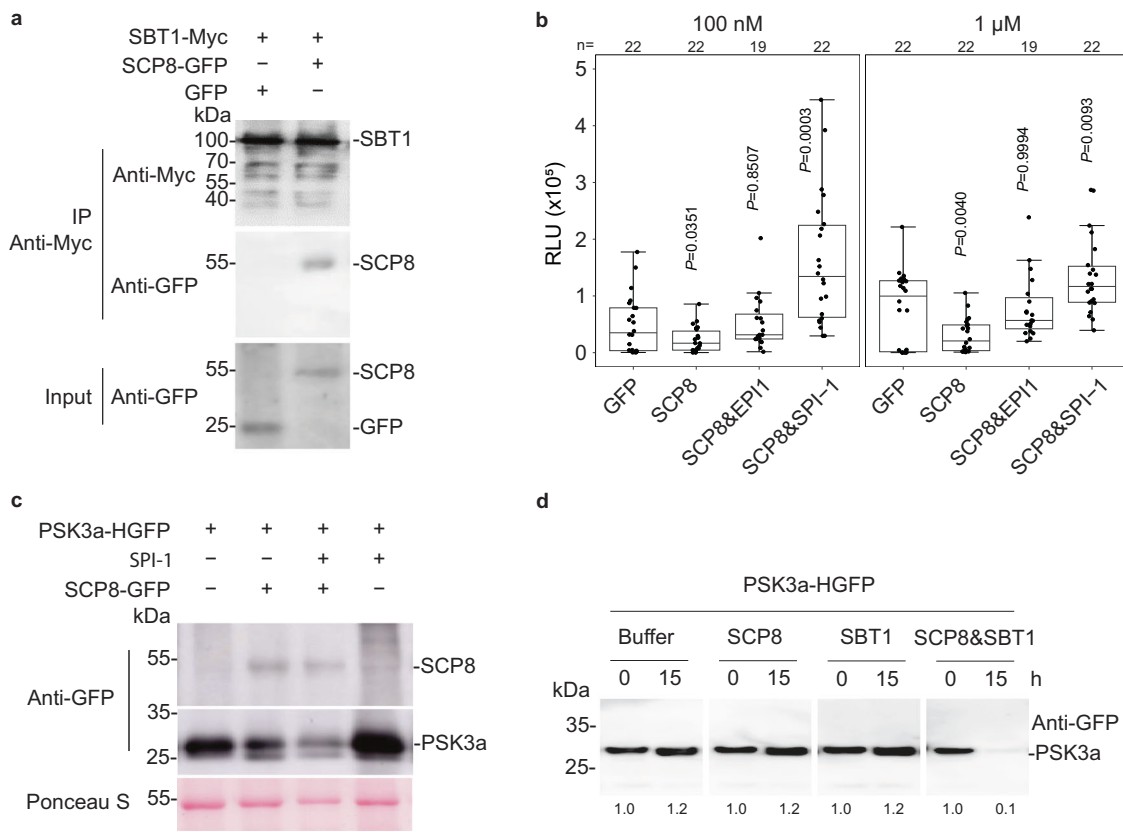

**Fig. 5 | Digestion of the *N. benthamiana* PSK3a precursor by SCP8. a** Co-immunoprecipitation and protein blot analysis of proteins from *N. benthamiana* leaves transiently co-expressing GFP-tagged SCP8 (SCP8-GFP) and myc-tagged *N. benthamiana* SBT1 (SBT1-myc). Total protein extracts from *N. benthamiana* leaves were collected 2 days post agroinfiltration and subjected to immunoprecipitation using anti-GFP agarose beads. Both total proteins (Input) and immunoprecipitated proteins (IP-myc) were analyzed via protein blot using anti-GFP and anti-myc antibodies. **b** ROS burst in *N. benthamiana* leaves transiently expressing GFP, SCP8, SCP8&EPI1, and SCP8&SPI-1, respectively, in response to various concentrations of flg22. Data points are indicated as black dots from three independent experiments and plotted as box plots (center line, median; bounds of box, the first and the third quartiles; whiskers, 1.5 x IQR; error bar, minima and maxima). Statistically differences between GFP-expressing plants and plants expressing the indicated proteins were analysed using a two-sided Student's *t*test. **c** Protein blot analysis of apoplastic

fluids isolated from *N. benthamiana* leaves expressing PSK3a-HGFP alone, co-expressing either SPI-1 and SCP8 individually, or both SPI-1 and SCP8. Protein detection was performed using an anti-GFP antibody. Approximate PSK3a-HGFP protein levels were determined by measuring band intensities from western blots using ImageJ software. Band intensities were normalized to the corresponding loading controls. The protein level in the first lane was set to 1.0 for normalization. The experiment was repeated 3 times with similar results. **d** PSK3a-HGFP purified from *N. benthamiana* was incubated at room temperature for 0–15 h in reaction buffer containing SCP8, SBT1, or both SCP8 and SBT1. Proteins mixtures were analyzed using an anti-GFP antibody. Approximate protein levels were determined by measuring band intensities from western blots using ImageJ software. For each condition, the band intensity at the 0 time point was set to 1.0 and the relative intensities were indicated below the blot. The experiment was repeated 3 times with similar results.

## Discussion

PSK-PSKR1 signaling has previously been shown to negatively regulate plant innate immunity[11,51,53]. Activation of PSK signaling in Arabidopsis enhances plant susceptibility to biotrophic and hemi-biotrophic pathogens, such as *Pseudomonas syringae*, *Ralstonia solanacearum*, *Fusarium oxysporum*, and *Hyaloperonospora arabidopsidis*[11,51,53,54]. This increased susceptibility is likely due to the recruitment of the PRR co-receptor BAK1, which restricts its availability for the formation of PRR-BAK1 complexes necessary for PTI activation[16,27]. Recent studies have shown that PSK signaling negatively regulates salicylic acid (SA) signaling[53]. Conversely, PSK signaling positively regulates auxin signaling, which is crucial for plant resistance to necrotrophic pathogens, including *Alternaria brassicicola* and *Botytis cinerea*[54,55].

As a hemi-biotrophic soil-borne fungal pathogen, *Verticillium dahliae* exhibits prolonged host colonization, bypassing the biotrophic phase during early infection and adopting a necrotrophic phase in later stages. Therefore, PSK treatment or overexpression of PSK genes in plants could result in varying responses to *V. dahliae* depending on the timing, dose, and method of treatments[56].

The *V. dahliae* genome encodes RALF-like peptides that suppress PTI; however, no homologous PSK genes have been found in any fungal or pathogen genomes. Unlike RALF genes, which are expressed ubiquitously in different plant tissues, PSK genes are primarily expressed in vascular tissues. In these tissues, both host and pathogens secrete molecules to manipulate each other. Previous studies have shown that host plants secrete small RNAs targeting virulence genes of vascular-colonizing *V. dahliae*[57]. In this study, we demonstrate that a small cysteine-containing protein, SCP8, secreted by *V. dahliae*, manipulates PTI activation by enhancing PSK signaling (Fig. 6e).

SCP8 is one of the few known microbial effectors that target a plant apoplastic component to suppress PTI[1,58]. SCP8 forms complexes with plant subtilases, a group of serine proteases involved in processing hormone peptide precursors, including RALFs and PSKs[46]. This interaction suggests that SCP8 likely influences PSK precursor digestion by forming complexes with various plant subtilases, leading to differential PTI suppression across different plant species.

Co-expression of SCP8 with subtilase inhibitors eliminated the suppression of PTI in *N. benthamiana*, suggesting that the accumulation of PSK induced by SCP8 was inhibited. However, the GFP-tagged

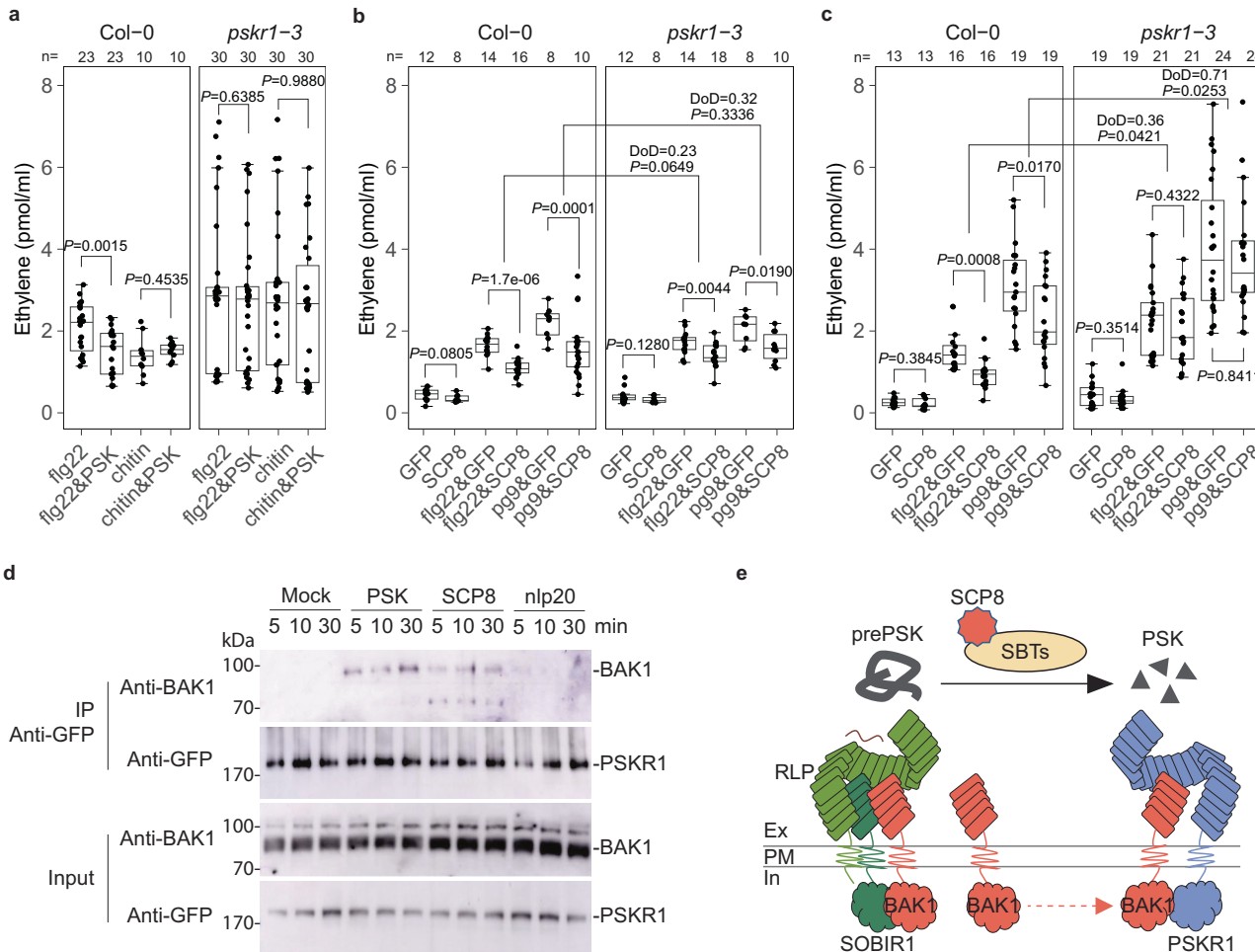

**Fig. 6 | PTI-suppression and PSK signaling activation in the presence of SCP8 protein. a** Ethylene production in response to flg22 and chitin. Ethylene levels were measured in Col-0 and *pskr1-3* mutant plants 4 h after infiltration with flg22 (1 μM), a combination of flg22 and PSK (both at 1 μM), chitin (1 μM), and a combination of chitin and PSK (both at 1 μM). **b** Ethylene production in response to apoplastic fluids: Col-0 and *pskr1-3* plants were infiltrated with apoplastic fluids from *N. benthamiana* containing GFP or SCP8 (referred to as GFP or SCP8, respectively), and the ethylene production was measured 4 h after infiltration with water (control), GFP, SCP8, GFP with flg22 (1 μM), GFP with pg9 (1 μM), SCP8 with flg22 (1 μM), and SCP8 with pg9 (1 μM). The difference of differences (DoD) represents the difference between the mean change in ethylene production upon treatment with GFP or SCP8 in Col-0 and the *pskr1-3* mutant. The Welch's *t*test *P*value is shown below the DoD value. **c** Ethylene production in response to co-infiltration of proteins with flg22 or pg9: Col-0 and *pskr1-3* plants were co-infiltrated with GFP or SCP8 protein, along with either flg22 (1 μM) or pg9 (1 μM). Ethylene production was measured 4 h post-infiltration. For **a**, **b**, and **c**, data points are indicated as black dots from two-four independent experiments and plotted as box plots (center line, median;

bounds of box, the first and the third quartiles; whiskers, 1.5 x IQR; error bar, minima and maxima). Statistical differences between treatments with elicitor alone and elicitor plus PSK were analysed using a two-sided Student's *t*test. **d** Protein blot analysis of co-immunoprecipitated proteins from Arabidopsis plants expressing PSKR1-GFP. Plants were infiltrated with water (control), PSK (1 μM), SCP8 (1 μM) or nlp20 (1 μM). Total protein extracts (Input) and samples precipitated using GFP-trap beads were analyzed by protein blotting, using anti-GFP and anti-BAK1 antibodies to detect PSKR1-GFP and its interaction with BAK1. Statistically significant differences (*P* < 0.05, student's *t*test) are indicated by different letters. Band intensities were measured by ImageJ and the protein level in the first lane of each blot was set to 1.0 for normalization. The experiment was repeated 3 times with similar results. **e** SCP8 mod**e** of action *in planta*. SCP8 interacts with plant subtilases, promoting the cleavage of PSK precursors, which leads to the accumulation of PSK in the apoplastic space. PSK then binds to its receptor, PSKR1, activating downstream signaling by recruiting co-receptors such as BAK1 and BKK1. These co-receptors are also involved in plant immune signaling initiated by LRR-domain-containing cell surface immune receptors, such as RLP23 and RLP42.

PSK3a precursor remained susceptible to digestion by SCP8 even in the presence of subtilase inhibitors. This suggests that additional signaling pathways, beyond PSK signaling, may be altered to compensate for PTI suppression. These findings imply that SCP8 could form complexes with various subtilases, as supported by supplementary information shown in Supplementary Fig. 4b.

The digestion of the PSK3a precursor is highly dependent on the complex formation of SCP8 and *N. benthamiana* subtilase SBT1 (Fig. 5d). The exact mechanism by which the SCP8-subtilase complex promotes PSK precursor digestion remains unclear. Some SBTs can switch from an auto-inhibited to an active state[44]. Interaction with SCP8 could induce a conformational change in SBT1 that relieves this auto-inhibition. Alternatively, SCP8 may interfere with the binding of SPI-1-like inhibitors to

SBT1, thereby enabling its proteolytic activity toward PSK3a. The interaction of SCP8 with SBTs is not restricted to SBT1. It is conceivable that pathogen effectors like SCP8 utilize diverse plant subtilases to adapt to changing conditions[1,59]. Similarly, plant subtilases generate various endogenous peptides that regulate both development and stress tolerance[46,48]. Overactivation of certain subtilases can induce programmed cell death[60], which might explain our failure to obtain transgenic SCP8 lines from PSK receptor mutants in Arabidopsis, as auto-immunity is already detected in the single mutant *pskr1-3*[53].

SCP8 exemplifies how effector-mediated manipulation of a negative immunoregulatory phytocytokine contributes to a bewildering array of immunosuppressive infection strategies of plant microbial pathogens. Further research into the protease substrates and

specificities of SCP8 will enhance our understanding of the interplay between plant development and immunity and how plants and microbes adapt to environmental changes.

## Methods

### Fungal strains and transformation

The wild-type *Verticillium dahliae* (*Vd*) strain 592 (V592)[61] isolated from cotton was used as the recipient strain for DNA transformation. *Agrobacterium tumefaciens* strain EHA105 was used for *Vd* transformation. Potato dextrose agar (PDA) medium with or without antibiotics was used to culture V592 and transformants as previously described. The gene knockout mutants were selected by 50 μg/ml hygromycin B (Sigma-Aldrich), and gene complementation transformants were selected by 100 μg/ml sulfonylurea (Sigma-Aldrich)[42].

### Plant infection assays

Infection assays on cotton cultivars with *Vd* were performed according to Gao et al.[61]. Cotton disease quantification followed the method of Wu et al.[34]. For Arabidopsis and *N. benthamiana*, 7-10-day old seedlings germinated from soil were released by watering and immersed in *Vd* spore solution (approx. $10^7$/ml) for 10 s. Inoculated Arabidopsis seedlings were reburied into the soil and grown for 20 days at 22 °C and 80% humidity for phenotyping. Inoculated *N. benthamiana* seedlings were water-incubated for 20 days at 22 °C and 80% humidity for phenotyping. Fungal biomass quantification was conducted as described by de Jonge et al.[62].

### RNA sampling for RNAseq

*N. benthamiana* seedlings germinated in soil for 7 days were inoculated with two different *Vd* strains, V592 and V171, and incubated in water at 22 °C and 80% humidity. Non-inoculated seedlings and seedlings inoculated at 3, 6, 9, and 12 days were harvested for RNA isolation and sequencing. RNAseq was done by BGI (Shenzhen, China) and analyzed using *N. benthamiana* genome database (https://solgenomics.net/organism/Nicotiana_benthamiana/genome).

### Genes cloning and construct design

Gene sequences were obtained from genome databases of *Verticillium dahliae*, *Arabidopsis thaliana* and *N. benthamiana*[63–65] (Supplementary Table 2). Gene cloning primers were listed in Supplementary Table 4. The *SCP8* gene knockout construct was generated using the strategy described by Wang et al.[42] with primers listed in Supplementary Table 4. The *VdSCP8* gene construct for genetic complementation was assembled as described[32] with primers listed in Supplementary Table 4. For the *SCP8-GFP* fusion construct, the OliC promoter and open reading frames of *SCP8* and *GFP* were amplified by PCR (primers in Supplementary Table 4), and then joined by overlap extension PCR with primers PoliC_pSul_F and GFP_pSulC_R (Supplementary Table 4). The product was cloned into pSul-RG#PB vector[42] digested with HindIII and EcoRI using ClonExpress II One-Step Cloning Kit (Vazyme, China). All the constructs for *in planta* expression were made with primers listed in Supplementary Table 4 following the method described by Zhang et al.[32].

*SCP8* and *GFP* gene was amplified by PCR with primers listed in Supplementary Table 4, and ligated into pPICZα vector digested by *EcoR*I and *Not*I restriction enzymes, yielding pPICZα::*SCP8* and pPICZα::*GFP* constructs. The constructs were transformed into *P. pastoris* GS115 strain, and transformants were selected on YPD medium containing 50 μg/ml zeocin.

### Arabidopsis transformation and selection

*Arabidopsis thaliana* Col-0 was transformed by *A. tumefaciens* GV3101 strain expressing *pLOCGex::SCP8,* and T0 seeds were selected on ½ MS plates containing 50 μg/ml kanamycin.

### Transient expression in *Nicotiana benthamiana*

*A. tumefaciens* strains were grown overnight in YEB medium (0.5% (w/v) beef extract, 0.5% (w/v) bacteriological peptone, 0.5% (w/v) sucrose, 0.1% (w/v) yeast extract, 2 mM $MgSO_4$) with appropriate antibiotics and 20 μM acetosyringone at 28 °C. Cultures were harvested and resuspended in 2.0% (w/v) sucrose, 0.5% (w/v) Murashige and Skoog salts without vitamins, 0.2% (w/v) MES, and 0.2 mM acetosyringone, pH 5.6, to an optical density at 600 nm ($OD_{600}$) = 1.0. For co-expression, two cultures carrying appropriate constructs were mixed in a 1:1 ratio to $OD_{600}$ = 0.5 for each. The resuspended cultures were incubated at room temperature for 1–3 h, and infiltrated into 5-6-week-old *N. benthamiana* leaves with a 1 ml-syringe. Samples were collected 1, 2, or 3 days after agro-infiltration for gene expression analysis, microscopic analysis, or immunoblotting analysis. Agro-infiltration was performed 3 times for each analysis.

### RNA extraction and RT-qPCR

Total RNA was isolated using a Nucleospin® RNA Plant Kit (Macherey-Nagel, Germany) according to the manufacturer's instructions. First-strand cDNA was synthesized from 1 μg of total RNA with SuperScript® III Reverse Transcriptase (Invitrogen), according to the manufacturer's instructions. RT-qPCR was performed using a C1000 Thermal Cycler (Bio-Rad) in combination with the EvaGreen 2x qPCR MasterMix−iCycler (Applied Biological Materials). The primers used to detect the transcripts are listed in Supplementary Table 1. The RT-qPCR conditions were as follows: an initial 95 °C denaturation step for 3 min, followed by denaturation for 15 s at 95 °C, annealing for 20 s at 60 °C, and extension for 20 s at 72 °C for 40 cycles. The data were collected using the Bio-Rad CFX Manager software and processed in Microsoft Excel. The transcript levels of target genes were normalized to the transcript levels of *EF1α* gene either from Arabidopsis or *N. benthamiana*[27] (Supplementary Table 4) according to the $2^{-\Delta\Delta Ct}$ method[66].

### Elicitors and immune response measurements

Chitin (IsoSep, No.57/12-0010, MW 1643.57) was dissolved in water to make 100 μM stock. flg22, nlp20, and pg9 were synthesized by Genscript and dissolved in water to make 100 μM stocks. Elicitor induced immune responses, including ROS burst, MAPK activation, and ethylene production were measured. The ROS production was monitored from 0 to 60 min after elicitation by using Mithras LB940 Multimode Reader (Berthold Technologies), and the peak values were selected for statistical analysis. The ethylene production was measured from 2 to 4 h post-elicitation depending on experimental conditions by using Gas Chromatography GC-14A system (Shimadzu)[21]. For *Agrobacterium*-infiltrated *N. benthamiana* leaves, ethylene levels were measured 2 h after elicitation. For *Arabidopsis* Col-0 and SCP8 transgenic lines, measurements were taken 3 h after elicitation. To assess ethylene production after infiltration, apoplastic fluids, purified SCP8 protein, nlp20, pg9, or phytosulfokine (PSK) were either infiltrated individually or co-infiltrated into leaves, which were subsequently cut into small pieces and incubated for 2 h before ethylene quantification. For assays involving chitin or flg22, these elicitors were infiltrated either alone or together with PSK, apoplastic fluids, or SCP8 protein; leaves were then cut into pieces, incubated for 4 h, and assessed for ethylene production. To detect the MAPK activation, flg22- or chitin-treated protein samples were collected at 5 min post-elicitation, while nlp20- or pg9-treated protein samples were collected at 10 min post-elicitation. All protein samples were detected by Phospho-p44/42 MAPK antibody (Cell Signaling Technology)[21].

### Co-immunoprecipitation (Co-IP) and bimolecular fluorescence complementation (BiFC) assays

For Co-IP assay, about five-week old *N. benthamiana* leaves or six-week old Arabidopsis leaves were infiltrated either with agrobacteria for protein expressions or with purified proteins or synthetic peptides.

Total proteins were isolated from *N. benthamiana* leaves two days after agro-infiltration or from Arabidopsis leaves 5–30 min after protein infiltration in RIPA buffer (Thermo Fisher Scientific) mixed with protease inhibitors cocktail (Roche). Isolated proteins were mixed with anti-GFP or anti-myc agarose beads (Chromotek) and incubated at 4 °C for one hour followed by washing off unspecific bound proteins according to the product manual. Immunoprecipitated proteins were detected by western blot with anti-GFP (Torrey Pines Biolabs), anti-HA (Cell Signaling Technology), or anti-myc antibodies (Sigma-Aldrich).

For BiFC assay, *A. tumefaciens* strains carrying BiFC constructs were co-infiltrated into five-week-old *N. benthamiana* leaves. Constructs included combinations of SBT1 & SCP8, SBT1 & SBT1, and SCP8 & SCP8. Fluorescence was detected two days post-infiltration using a confocal laser scanning microscope (LSM880, Zeiss) with a ×63 water-immersion objective. Positive (RLP23 & SOBIR1) and negative (EFR & SOBIR1) controls were included as described previously[7,21]. Yellow fluorescent protein (YFP) was excited with a 514 nm laser, and emission was collected between 516-556 nm. Images were processed using ZENblue software (Zeiss) with standardized adjustments for brightness and contrast according to the method of Pruitt et al.[7].

### Mass spectrometry analysis

Total proteins were extracted from *N. benthamiana* leaves transiently expressing GFP (control, $n=1$) or SCP8-GFP ($n=1$). Protein extracts were filtered through a 0.45 μm membrane and immunoprecipitated using GFP-Trap agarose beads (Chromotek). Bound proteins were eluted and digested with trypsin (Promega) before LC-MS/MS analysis. Tryptic peptides were analyzed using an EASY-nLC 1000 system coupled via a Nanospray Flex ion source to an Orbitrap Fusion Tribrid mass spectrometer (Thermo Fisher Scientific). Peptides were loaded onto a trap column (Acclaim PepMap100 C18, 100 μm × 2 cm) and separated on an analytical column (Easy Column C18, 75 μm × 10 cm, 3 μm) at 300 nL/min. A 60 min gradient of solvent B (84% acetonitrile, 0.1% formic acid) in solvent A (0.1% formic acid) was applied as follows: 0–60% B for 50 min, 60–90% B for 4 min, and 90% B for 6 min. The mass spectrometer was operated in positive-ion mode using a data-dependent acquisition method selecting the top 20 precursor ions (m/z 300–1800) for HCD fragmentation. MS1 scans were acquired at a resolution of 70,000 (m/z 100) with AGC target $3 \times 10^6$ and 50 ms injection time; MS2 scans at 17,500 (m/z 100) with AGC target $1 \times 10^5$, 100 ms injection time, and 30 s dynamic exclusion. Spectra were searched with MASCOT (v2.2) against *N. benthamiana* and *N. tabacum* databases (https://solgenomics.sgn.cornell.edu/). Search parameters included trypsin digestion (≤2 missed cleavages), fixed carbamidomethyl (C), variable oxidation (M), precursor tolerance 10 ppm, and fragment tolerance 0.8 Da. Peptides (≥7 amino acids) were filtered to <1% FDR at peptide and protein levels. Proteins were accepted with ≥1 unique peptide and $P < 0.05$. Common proteins in GFP (MS-GFP) and SCP8-GFP (MS-SCP8) samples were removed as nonspecific interactors. The mass spectrometry proteomics data have been included in Source Data file.

### Protein expression, purification, and digestion assays

Proteins expressed in *Escherichia coli* were purified as described by Stührwohldt et al.[67]. Proteins expressed in *N. benthamiana* were purified following the method described by Reichardt et al.[48]. For PSK3a precursor expression and purification, both PSK3a and SPI-1 constructs were co-infiltrated into *N. benthamiana* leaves. For proteins express in *Pichia pastoris*, a single colony of SCP8 or GFP transformant was inoculated into YPD medium, and protein expression was induced by methanol in BMMY medium, as described in the *Pichia* protein expression handbook (Invitrogen). SCP8 protein was purified using affinity chromatography with a His-trap column (Cytiva), followed by size exclusion chromatography with a Superdex™ 200 column (Cytiva). PSK precursor digestion assays were performed according to Stührwohldt et al.[67]. For digestion assays, approximately 10 μg of

PSK3a precursor was incubated with either 100 ng of SBT1, 1 μg of SCP8, or a mixture of 100 ng SBT1 and 1 μg SCP8 at room temperature. All purified proteins were buffer-exchanged into 50 mM sodium phosphate buffer (pH 7.0) supplemented with 10 mM NaCl using ultrafiltration. Reactions were sampled at 1 min (defined as time zero) and up to 15 h. Digestion products were mixed with 3× SDS loading buffer, denatured at 95 °C for 5 min, and stored at −20 °C until analysis by SDS−PAGE and immunoblotting with anti-GFP, anti-His$_6$ or anti-myc antibodies.

### Reporting summary

Further information on research design is available in the Nature Portfolio Reporting Summary linked to this article.

## Data availability

The RNAseq data generated in this study have been deposited in the NCBI database under accession code GSE236729. The protein mass spectrometry data are provided in the Source Data file. Source data are provided with this paper.

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

## Acknowledgements

This work was supported by Deutsche Forschungsgemeinschaft (DFG) grant Nu70/17-1 to T.N., the National Natural Science Foundation of China (No. 32020103003) to H.-S.G, and National Natural Science Foundation of China (No. 31500119) to C.H. We thank Dr. Birgit Kemmerling, Prof. Sebastian Wolf, Dr. Harald Keller, and Prof. Klaus Harter for providing Arabidopsis *pskr1-3* and *pskr1-3pskr2* mutant lines, PSK gene overexpressing lines, PSKR1-GFP construct and transgenic line for our study. We also thank Dr. Yu Zhou and Dr. Ying Li for technical support with BiFC assays.

## Author contributions

C.H., H.-S.G., and T.N. conceived and designed major experiments; A.St., A.S. and C.H. designed the experiments with protease activity assays. L.Z. and C.H. conducted experiments; C.H. and T.N. wrote the manuscript. All authors analyzed data, discussed the results and commented on the manuscript.

## Funding

## Competing interests

The authors declare no competing interests.
