## [Transparent Peer Review file · Nature Communications]

A fungal pathogen manipulates phytoytokine signaling for plant infection

Corresponding Author: Professor Thorsten Nuernberger

Version 0:

Reviewer comments:

Reviewer #1

(Remarks to the Author)

The manuscript by Hua et al. describes the characterization of a secreted effector from *Verticillium dahliae* (Vd), SCP8, which is very important for Vd virulence. The paper provides some evidence about the molecular mechanism of SCP8 function. SCP8 localizes to the apoplast (previously shown) and promotes PSK phytoytokine signalling to negatively regulate PTI. SCP8 may interact with subtilases to promote PSK production. Overall, the paper gives some interesting and novel insights into pathogen strategies to manipulate immunity (hijacking of an endogenous phytoytokine pathway), but some important details remain unclear and require further experimentation. Detailed issues are listed below:

Major issues:

- The authors suggest that Vd effectors may promote PSK/PSKR1/PSKR2 expression in *N. benthamiana*. To support this hypothesis, the authors should test whether MAMP perception is insufficient to induce expression of the respective genes.
- The exact effect of SCP8 on LRR-RK/RLP-mediated PTI signaling pathways remains a bit unclear. The flg22 response seems to be strongly inhibited in *N. benthamiana*, but not upon overexpression in *Arabidopsis*. What is the author's explanation for this? Is it due to the lower levels of overexpression in *Arabidopsis*? The overexpression lines were also not characterized. It remains unclear how strong the overexpression in the two lines used for experiments in the paper is.
- It is interesting that the authors state that it was impossible to obtain SCP8-expressing transgenic lines in the *pskr1*, *pskr1/pskr2* and *pskr1/psy1r* mutant backgrounds. However, it is very strange that the constant presence of the effector is only detrimental in these receptor mutants. This argues against the fact that PSK-PSKR1 signaling is the primary target of SCP8, since one would expect a reduced impact on overall plant growth in the receptor mutants. This needs appropriate documentation and discussion.
- PSKR1 is the predominant PSK receptor in *Arabidopsis*, but can function partially redundant with PSKR2 (Amano et al., 2007). What is the contribution of PSKR2 to *Verticillium* resistance (Fig. 1) and SCP8 sensitivity (Fig. 4). This is particularly relevant, since SCP8 still has a significant inhibitory effect on flg22 and pg9-triggered ethylene accumulation (Fig. 4B)
- Fig. 4C: The data is hard to interpret, as a control is missing. The authors need to compare flg22 and pg9-elicited ethylene accumulation upon co-infiltration of a negative control, not only water (e.g. GFP-6xHis).
- The authors report the identification of the SBT Niben101Scf00726g02006.1 as an interactor of SCP8 and potential virulence target. They propose that SCP8 promotes SBT-mediated release of PSK from precursor propeptides to promote Vd virulence. However, to justify this conclusions, the authors should do some more experiments: Is NbSBT1-myc able to cleave NbPSK3a, e.g., or other PSK peptide precursors in vitro? Does SCP8 enhance activity of NbSBT1 in vitro? Can this activity be blocked by e.g. EPI1? Can SCP8 interact with *Arabidopsis* SBTs as well? What is the effect of SBT mutations on Vd virulence? The authors should attempt to see whether SBT1 homologous genes in *Arabidopsis* are important for Vd susceptibility/resistance, which may be difficult to the redundancy of the gene family.
- The *dscp8* *Verticillium* mutants have a striking loss of virulence. The authors suggest, that this mutation does not affect overall Vd growth on plates. However, the only proxy the authors present is colony diameter. They should also include a picture of fungal growth and the ability to sporulate on plates comparing the WT with the *scp8* mutants/complementation lines.

Minor issues:

- All bar graphs should be replaced by box/dot blots to visualize individual data points (e.g. ethylene data in Fig. 4)
- The legends miss the information about how many replicates were performed and the n of depicted data points.
- The western blot in Fig. 3C needs a chitin treatment control, to show that different lines (in particular T-12) are still able to activate MAPKs
- Fig. S1D lacks characterization of the native-VdSCP8 complementation line

- Is it possible that Vd produces PSK by itself? The authors should discuss this possibility.
- Fig. 4: The CoIP lacks the 30 min water control. Also, why does SCP8 not precipitate BAK1 at 30min post treatment, while the PSK effect is unaltered at the different time points analysed?
- Since the authors generated the dscp8 mutants and SCP8-GFP complementation lines, it would be nice to visualize extracellular secretion of SCP8-GFP upon infection in vivo by confocal microscopy.
- Why do the authors sometimes use PSK3 and sometimes PSK6 to test for expression of PSK peptide precursor genes?
- The pskr1 psy1r double mutant is not only affected in PSK signalling, but also PSY1. This should be rephrased

Reviewer #2

(Remarks to the Author)

The study demonstrates that effector VdSCP8 from *Verticillium dahliae* manipulates plant immune responses by modulating the PSK signaling pathway. The author found that SCP8 interacts with the subtilase protein SBT1, suggesting that SCP8 suppresses plant defenses by enhancing subtilase activity and generating more PSK. It asserts that PSK acts as an immunosuppressive factor. However, this contradicts some existing literature reports. In previous work, it is reported that *Arabidopsis* PSKR1 modify the immunity to biotrophic and necrotrophic pathogens in an antagonistic manner, PSK signalling via AtPSKR1 contributes to resistance against the fungal necrotrophic (Igarashi et al., Plant J 2012; Mosher et al., Plant J 2013). By contrast, PSK signalling attenuate the PAMP response to the bacterial pathogen (Loivamakia et al., 2010 *Physiologia Plantarum*; Igarashi et al., Plant J 2012; Mosher et al., Plant J 2013). This phenomenon has also been observed in other species. For example, in tomato, PSK and its receptor PSKR1 positively regulate resistance to fungal *Botrytis cinerea* (Zhang et al., Plant Cell 2018), but negatively regulate resistance to bacterial *Pst* DC3000 (Ding et al., EMBO J 2023). In cotton, PSK increases the resistance of cotton to *V. dahliae* (Zhang et al., Plant Physiol 2022). Therefore, PSK generally enhances the resistance to fungal diseases, which contradicts the author's perspective that PSK acts as an immunosuppressive factor. The contradictions need to be discussed and explained. Moreover, the conclusions drawn in the manuscript require more robust experimental evidence for substantiation. Overall, the study falls short of the standard required for publication.

Major issues:

- 1) As I mentioned above, the claim that PSK signaling acts as an immunosuppressive factor is not appropriate, and further study is needed to figure out the role of PSK in the resistance to *V. dahliae* of *Arabidopsis*, tobacco, and cotton.
- 2) The evidence for protein-protein interaction between SCP8 and SBT1 is not solid. The interaction is supported by only one co-immunoprecipitation (Co-IP) experiment, and there is no experimental evidence to confirm whether this interaction occurs in the apoplast. Furthermore, whether SBT1 is involved in the synthesis of PSK remains uncertain. The SBT family consists of multiple members that play roles in the synthesis of various peptides. Therefore, it is unclear if SBT1 specifically participates in PSK synthesis. Moreover, SBT inhibitors are likely to affect synthesizing multiple peptides, making it difficult to attribute the changes in resistance solely to the impact on PSK synthesis. Other peptide synthesis processes may also be affected, which could contribute to the observed resistance changes.
- 3) In Fig 3E, co-expression of SCP8 with SBT inhibitor SPI-1 reduced SCP8-mediated defense suppression. However, why co-expression of SCP8+SPI-1 triggers a higher ROS response than the control?
- 3) In Fig 4A, it has been observed that PSK can inhibit the resistance induced by flg22, a bacterial flagellin-derived peptide. However, PSK fails to inhibit the resistance response triggered by C8, indicating a potential differential role of PSK in regulating resistance against bacterial and fungal pathogens.
- 5) This study focuses on the fungal pathogen *V. dahliae*. Why did the authors use the bacterial flagellin protein flg22 as a PAMP to investigate a fungal pathogen manipulating plant infection?
- 6) The authors did not provide specific information about the number of experimental replicates conducted or biological replicates within each experiment. Additionally, some experiments, such as the transcriptomic data in Fig 1A, and Fig 1G, appear to have only two biological replicates, which is a limited sample size. It is essential to have an adequate number of replicates to account for biological variability and ensure the statistical significance of the findings. The inadequate number of biological replicates would raise doubts about the reliability and robustness of the experimental results.
- 7) The writing style of this manuscript needs to be improved. Currently, Introduction and Results sections are somewhat mixed together, a clear demarcation is necessary. And the Discussion section is missing. A discussion section allows researchers to compare their findings with previous studies, identify potential limitations or discrepancies, propose explanations, and generate new research hypotheses.

Minor issues:

- 1) Why was the *Arabidopsis* PR1 signal peptide used, and what is PR1? It should be explained in the manuscript.
- 2) When was the disease condition observed and recorded? In line 131, it is stated as 30 days, while in line 425, it is mentioned as 25 days. Please clarify.
- 3) In line 141, it is not specified which genes the transgenic lines T-6 and T-12 carry. It is only in line 149 that SCP transgenic lines are mentioned. It is necessary to clarify in line 141 the specific gene types to avoid any confusion for readers.
- 4) The time point of ROS burst and MAPK activity measurement should be mentioned in the manuscript.

- 5) In line 174, please note that the term "restore" is inappropriate, since the *pskr1* mutant exhibits significantly lower levels of ethylene content.
- 6) In Fig 4b, SCP-containing apoplastic fluid and elicitor were treated together. While in Fig 4C, the treatments of SCP8 protein were prior to flg22 treatments. Why were SCP8 protein and elicitor treated differently in two similar experiments?
- 7) Why does the band intensity of GFP after immunoprecipitation (IP) in Figure 4D decrease over time?
- 8) The method for pathogen inoculation needs to be described in more detail. The cotton planting conditions and RNA-seq were not mentioned in the Method.
- 9) The working model requires further clarification. What does different colored triangles mean? The meaning of gray and purple color also needs to be explained. Additionally, the figure legend of the working model needs to be more detailed.

Reviewer #3

(Remarks to the Author)

It is an interesting story that a pathogen promotes a growth signaling to interfere with immune signaling. However, there are multiple points that need to be addressed in experimental results before this work is properly evaluated.

In general, the results without statistics, such as Figs 3CD and 4D, need to be associated with a statement like, "Similar observations were made in XX independent experiments." And such replicated results need to be publicly shared (supplemental figures, Figshare, or etc.? – the method of sharing should be discussed with the editorial office). In addition, for the results with statistics, it needs to be clarified whether the replicate values are from independent experiments.

Figs 3AB. (1) Each of ethylene and ROS measurements, a full set of molecular patterns, flg22, nlp20, pg9, and C8 should be shown. Similarly, Fig 3C should include C8.

The MPK phosphorylation is clearly lower in T-6 and T-12 after flg22 treatment (Fig 3C). However, the ethylene response is not (Fig 3A, flg22 panel). This point should be discussed. It is OK to say that the reason is unknown – it just needs to be clearly pointed out to readers that there is some observation inconsistent with a simple hypothesis.

Fig 4A. The y-axis scales of two plots (Col-0 and *pskr1-3*) should be the same for an easier comparison. Then it becomes evident that the value for flg22 (with no PSK) in Col-0 is about three times higher than that for flg22 (with no PSK) in *pskr1-3*. This trend is also clear relative to the values for C8. Thus, flg22 response is generally reduced in *pskr1-3*. And due to this flg22 response difference in two genotypes, the effect of PSK on flg22 response cannot be evaluated.

Fig 4B. An appropriate statistical comparison needs to be performed: difference in differences. For each of flg22 and pg9 treatments, whether the value $\{(GFP_Col - SCP8_Col) - (GFP_pskr - SCP8_pskr)\}$ is significantly larger than 0 needs to be statistically evaluated.

There should be a discussion on the mode of action of SCP8, whether the effect of SCP8 is directly on PSKR1-mediated signaling, such as SCP8 as a ligand for PSKR1, or rather indirect, such as somehow SCP8 induces the PSK expression and the higher PSK concentration turns on PSKR1-mediated signaling. The rapid PSKR1-BAK1 association induction 5 min after SCP treatment (Fig 4D) strongly suggests the former case. A response within 5 min in plants is unlikely caused via accumulation of the encoded protein after transcriptional induction of a gene. It is confusing regarding this point as the story starts with transcriptional induction of the *SOBIR1*, *PSKR1*, and *PSK* genes in much slower kinetics (days; Fig 1A).

Version 1:

Reviewer comments:

Reviewer #1

(Remarks to the Author)

The authors addressed most concerns raised during the revision and significantly improved the paper. However, some unclarity remain that are listed below and should be addressed before publication of the article.

- Figure S2C: Optimized Ethylene assay in Arabidopsis upon infiltration lacks statistics
- Line 136: When describing this optimised assay, please state in the text what is different to the treatment presented in Fig. 4A
- Fig. S4 is quite confusing. Please clarify the points listed here: 1) What is the second lane of the western blot? Is this the control lane without SBTs? The top lane marks only SCP8-HA „+“, which is also reflected in the input. But what is the difference between lane 1 and lane 2, since SCP-HA is only ColIPed in lane 2, where there is actually no SBT present (neither in input, nor in IP). 2) Why is there no signal for AtSBT1.3 in the input lane?
- line 163 (The *PSK3a* gene,, was synthesized with a C-terminal His6-GFP tag...) sounds odd. The gene does not have a C-terminal His6-GFP tag. Please rephrase.
- Fig. S4B: The labelling should be corrected. It labels as SPIAT/NB, but I assume SPI-1 from different species is meant.
- Line 169: When describing results in Fig. S4B, the authors state „significant“ accumulation of the PSK precursors. Either provide quantification, or remove significant.
- Fig. S5C: The loading is very unequal in the different compared lanes, which likely affects the drawn conclusions. Please

provide a better blot result here instead.

- Fig. S6: Is there a labelling issue in this figure? The authors state that incubation of PSK3-GFP with a mixture of SBT1/SCP8 induces PSK3 cleavage after 3 hours, but there are no intensity changes in the GFP blot. There are some changes in the a-myc blot, which should however detect SBT1-Myc. Also, it would be good if the tag is indicated within the Figure or at least in the legend, which is currently not the case.

- In general, figure legends/labelling often lack fusion tags for proteins, which makes interpretation of the blots difficult, as one cannot directly see which fusion protein is detected with which antibody. This should be changed throughout the manuscript.

Reviewer #2

(Remarks to the Author)

The manuscript investigated the role of the fungal effector SCP8 from *Verticillium dahliae* in modulating plant immunity through the PSK-PSKR1 signaling pathway. While the study demonstrates some interesting findings, there are significant issues that need to be addressed to strengthen the conclusions and improve the clarity of the work. The evidence presented in some key areas is thin, which undermines the reliability of the results and the confidence in the overarching conclusions. Below are my detailed comments:

Major:

1. Use of flg22 instead of fungal PAMPs to assess SCP8-mediated immunity weakens the validity of the conclusions. The authors primarily use flg22 (a bacterial PAMP) to demonstrate that SCP8 suppresses PTI by disrupting the RLP/SOBIR1-BAK1 immune signaling complex via competition for BAK1 recruitment. However, SCP8 is a fungal effector, and its relevance to fungal PAMPs such as chitin (C8) is underexplored. In fact, the results suggest that chitin does not elicit a significant immune response in the experimental system, weakening the biological relevance of the study.

2. Cleavage of PSK3a by the SCP8-SBT1 complex lacks robust evidence.

In Fig. 5d, PSK3a does not degrade when incubated with SBT1 alone. However, this contradicts Fig. 5c, where inhibition of SBT1 by SPI-1 clearly prevents PSK3a cleavage. Does this suggest that SBT1 typically cannot cleave PSK3a without SCP8? Or is the degradation condition-dependent?

In Fig. 5c shows two bands for PSK3a (likely precursor and cleaved forms), but Fig. 5d shows only one band.

In Fig. 5c, the PSK3a band becomes weaker upon SPI-1 treatment, which seems counterintuitive given that SPI-1 should inhibit SBT activity.

Additionally, the input protein levels in Fig. 5d appear inconsistent, weakening comparability.

3. Direct evidence of PSK's role in resistance to *V. dahliae* is missing. The study heavily relies on the PSKR1 receptor mutants to infer PSK's involvement in fungal resistance. However, direct evidence showing the role of PSK peptides themselves in resistance is missing. Apply exogenous PSK to the infection system to demonstrate its role in *V. dahliae* resistance or susceptibility would help.

4. Evidence for SCP8-SBT interaction is insufficient. Only one protein interaction method was used to demonstrate the interaction between SCP8 and SBT1. Additional approaches, such as yeast two-hybrid, BiFC, or other orthogonal methods, would strengthen this claim.

5. In fig. 6b and c, why co-infiltration of SCP8 leads to different results compared to experiments using apoplastic fluids containing SCP8?

Minor:

1. Please quantify protein levels in all relevant figures (e.g., Figs. 5c and 5d) to enable direct comparison.

2. In Fig. 1a, using raw reads number to represent gene expression is not appropriate as it fails to account for sequencing depth or gene length. Please Use TPM or FPKM normalization for expression analysis.

3. The variance analysis in Fig. 1a seems inconsistent with the earlier version of the manuscript. Ensure the statistical methodology is appropriate and results reproducible.

4. In Fig. 5c, why is the Ponceau S staining labeled as SCP8?

5. Line 82: Clarify how the conclusion that "VdSCP8 suppresses BAK1-dependent immune responses" is reached, as BAK1 is not directly discussed.

6. Line 111: Define "PEN" for better reader comprehension.

7. Please explicitly mention that chitin is referred to as C8 for clarity.

Reviewer #3

(Remarks to the Author)

Authors' response to point 1 of my previous review is satisfactory.

Authors' response to point 2 of my previous review:

Line 135. The method difference for Fig S2C needs to be specified instead of saying, "an optimized method". It seems that the ethylene level was measured after 4 hpi instead of 3 hpi for Fig 2. Actually, for Fig 2 line 134 says 3hpi, but line 330 says 2 hpi – make the method description consistent. If different experimental conditions are used for different figures, make the difference clear in the figure legends.

The chitin results should also be shown with the Fig S2C conditions, to clarify whether even with the "optimized method", no significant difference was observed with chitin.

Authors' response to point 3 of my previous review:

The authors' response tells that the original version had experiments performed under different conditions compared. I cannot believe that the authors did that.

Based on my previous comment, I think that the previous figure included C8. Why were the C8 results omitted in the new figure?

Authors' response to point 4 of my previous review:

I do not see the authors have corrected Fig 6b for the point. In the figure, it still looks that only the difference between +GFP and +SCP8 is tested separately for Col-0 and pskr1-3.

Minor points

1. Around line 79. Figs 1c and 1d need to be referred to.
2. Some long sentences could be benefitted by dividing them into short sentences. For example, the sentence starting line 76.

Version 2:

Reviewer comments:

Reviewer #1

(Remarks to the Author)

The authors addressed my recent concerns sufficiently. However, in response to Q6 of referee 2 (influence of PSK application on Vd infection success) the authors show data of the PSK4OE line in Ws-4. This data is however not in the paper. I strongly recommend to include this data since it supports the authors conclusion.

Reviewer #2

(Remarks to the Author)

The authors have made efforts to address my initial concerns; however, several issues still require further clarification or improvement:

1. Regarding my previous Q2, the authors suggest that SBT1 alone is not responsible for PSK3a cleavage, and that SCP8 from the pathogen enables SBT1 to cleave PSK3a. I was wondering how SCP8 confers this new function to SBT1. What is the molecular mechanism by which SBT1, which normally does not target PSK3a, gains this specificity upon interaction with SCP8?

2. In Fig. S3A, the BiFC panel "SBT1-nYFP & SBT1-cYFP (positive control for SBT1 homodimerization)" unexpectedly shows no fluorescence signal. Why is there no signal in this positive control? Additionally, in the SBT1-nYFP/SCP8-cYFP panel, the interaction also appears to occur around nuclear—indicated by a clear ring-like signal. Please clarify the subcellular localization and interpretation of this result.

3. In Fig. S3B, authors state that myc-tagged FLS2 was used as a negative control. However, a clear band is observed above 180 kDa in the IP-myc lane corresponding to FLS2, which appears to be a positive signal. Please clarify.

4. Figure 2b lacks statistical analysis

5. Quantification of protein levels is still insufficient, no quantification or labels are provided for other key figures such as Fig. 6d. Please revise accordingly. And the quantified values are not displayed directly above or below the target panel in figure 5C.

6. The explanation of how the authors conclude that "VdSCP8 suppresses BAK1-dependent immune responses" remains vague. Further clarification is necessary in the main text.

Reviewer #3

(Remarks to the Author)

Regarding Fig 6b. The P values for DoD tests are 0.336 and 0.0649. Thus, these two DoD values are NOT SIGNIFICANT. So, the conclusion that pskr1-3 attenuated the effect of SCP8 is NOT CORRECT. The effect sizes of DoD (0.32 and 0.23) do not tell the significance.

Version 3:

Reviewer comments:

Reviewer #2

(Remarks to the Author)

The authors have sufficiently addressed the issues I previously raised.

POINT-BY-POINT RESPONSE TO REVIEWER COMMENTS

Reviewer #1:

The manuscript by Hua et al. describes the characterization of a secreted effector from *Verticillium dahliae* (Vd), SCP8, which is very important for Vd virulence. The paper provides some evidence about the molecular mechanism of SCP8 function. SCP8 localizes to the apoplast (previously shown) and promotes PSK phytoytokine signalling to negatively regulate PTI. SCP8 may interact with subtilases to promote PSK production. Overall, the paper gives some interesting and novel insights into pathogen strategies to manipulate immunity (hijacking of an endogenous phytoytokine pathway), but some important details remain unclear and require further experimentation. Detailed issues are listed below:

Major issues:

1- *The authors suggest that Vd effectors may promote PSK/PSKR1/PSKR2 expression in N. benthamiana. To support this hypothesis, the authors should test whether MAMP perception is insufficient to induce expression of the respective genes.*

Our response: We thank the reviewer for their suggestion. The results have been incorporated into Figure 1b of our revised manuscript. However, we found that MAMP perception does not lead to upregulation of the PSK or PSKR genes.

2- *The exact effect of SCP8 on LRR-RK/RLP-mediated PTI signaling pathways remains a bit unclear. The flg22 response seems to be strongly inhibited in N. benthamiana, but not upon overexpression in Arabidopsis. What is the author's explanation for this? Is it due to the lower levels of overexpression in Arabidopsis? The overexpression lines were also not characterized. It remains unclear how strong the overexpression in the two lines used for experiments in the paper is.*

Our response: The results of SCP8 gene expression levels in transgenic *Arabidopsis* and transiently expressed *Nicotiana benthamiana* have been included in Figure S2 of our revised manuscript. Based on these results, SCP8 expression is not higher in *N. benthamiana* than in *Arabidopsis*. The observed differences in the extent of inhibition may be due to several factors, one of which could be the variation in host targets, such as subtilase types and PSK precursors, between the two plant species.

3- *It is interesting that the authors state that it was impossible to obtain SCP8-expressing transgenic lines in the pskr1, pskr1/pskr2 and pskr1/psy1r mutant backgrounds. However, it is very strange that the constant presence of the effector is only detrimental in these receptor mutants. This argues against the fact that PSK-PSKR1 signaling is the primary target of SCP8, since one would expect a reduced impact on overall plant growth in the receptor mutants. This needs appropriate documentation and discussion.*

Our response: Since PSK signaling negatively regulates plant immunity, the pskr1-3 single mutant or related double mutants may already be in a heightened state of immunity or autoimmunity, as recently reported by Song et al. (Nat. Plant, 2023). It is possible that overexpressing SCP8 further exacerbates this autoimmunity, leading to cell death in the pskr1 single or double mutants. Alternatively, because SCP8 forms complexes with multiple subtilases (Fig. S3), the activation of specific subtilases could enhance autoimmunity or promote cell death in these mutants. We have discussed these potential mechanisms in the Discussion section of our revised manuscript.

4- *PSKR1 is the predominant PSK receptor in Arabidopsis, but can function partially redundant with PSKR2 (Amano et al., 2007). What is the contribution of PSKR2 to Verticillium resistance (Fig. 1) and SCP8 sensitivity (Fig. 4). This is particularly relevant, since SCP8 still has a significant inhibitory effect on flg22 and pg9-triggered ethylene accumulation (Fig. 4B)*

Our response: In the updated manuscript, we show in Figures 1c and 1d that both the pskr1-3 single mutant and the pskr1-3/pskr2-1 double mutant are more susceptible to *V. dahliae* infection, suggesting that the contribution of PSKR2 to PSK signaling and resistance to *Vd* is minimal. This is consistent with the structural analysis of PSK-PSKR recognition, which indicates that PSKR1, but not PSKR2, functions as the primary receptor for PSK.

In Figure 4B (now Figure 6b in the updated manuscript), the remaining inhibitory effect could be attributed to factors present in the apoplastic fluid other than PSK, as SCP8 may either be targeting multiple subtilases or undergoing degradation.

5- *Fib. 4C: The data is hard to interpret, as a control is missing. The authors need to compare flg22 and pg9-elicited ethylene accumulation upon co-infiltration of a negative control, not only water (e.g. GFP-6xHis).*

Our response: The data have been updated to include the GFP-6xHis fusion protein as a control.

6- *The authors report the identification of the SBT Niben101Scf00726g02006.1 as an interactor of SCP8 and potential virulence target. They propose that SCP8 promotes SBT-mediated release of PSK from precursor propeptides to promote Vd virulence. However, to justify this conclusions, the authors should do some more experiments: Is NbSBT1-myc able to cleave NbPSK3a, e.g., or other PSK peptide precursors in vitro? Does SCP8 enhance activity of NbSBT1 in vitro? Can this activity be blocked by e.g. EPI1? Can SCP8 interact with Arabidopsis SBTs as well? What is the effect of SBT mutations on Vd virulence? The authors should attempt to see whether SBT1 homologous genes in Arabidopsis are important for Vd susceptibility/resistance, which may be difficult to the redundancy of the gene family.*

Our response: In the updated manuscript, we demonstrate that the *N. benthamiana* PSK3a precursor can be processed by the complex formed between SCP8 and *N. benthamiana* SBT1, but not by either SCP8 or SBT1 alone (Fig. 5). Additionally, our Co-IP results show that SCP8 forms complexes with subtilases from both *Arabidopsis* and *N. benthamiana*, specifically AtSBT1.3, AtSBT1.9, and NbSBT1 (Fig. S3).

As per Reviewer 1's suggestion, we tested the susceptibility of *Arabidopsis* sbt1 mutants to *V. dahliae* infection. While some sbt1 mutants (SBT1.1, SBT1.8) show increased resistance to *Vd* infection compared to the wild-type Col-0 (see figures below), we cannot yet determine which specific SBT genes are targeted by SCP8 based on the current data. Our future work will focus on elucidating the precise mode of interaction between SCP8 and plant subtilases. As this reviewer rightly points out, the complexity of the subtilase gene family and the potential functional redundancy among its members complicate these analyses.

7- *The vdsctp8 Verticillium mutants have a striking loss of virulence. The authors suggest, that this mutation does not affect overall Vd growth on plates. However, the only proxy the authors present is colony diameter.*

They should also include a picture of fungal growth and the ability to sporulate on plates comparing the WT with the scp8 mutants/complementation lines.

Our response: The data were previously reported by Wu et al. (Nat Commun, 2023), and we have cited this reference in our manuscript.

Minor issues:

1- All bar graphs should be replaced by box/dot blots to visualize individual data points (e.g. ethylene data in Fig. 4).

Our response: This has been addressed in the updated MS.

- The legends miss the information about how many replicates were performed and the n of depicted data points.

Our response: This information has been added to the MS.

2- The western blot in Fig. 3C needs a chitin treatment control, to show that different lines (in particular T-12) are still able to activate MAPKs

Our response: MAPK activation by chitin has been added to the MS (Fig 4c).

3- Fig. S1D lacks characterization of the native-VdSCP8 complementation line.

Our response: The data were previously reported by Wu et al. (Nat Commun, 2023), and we have cited this reference in our manuscript.

4- Is it possible that Vd produces PSK by itself? The authors should discuss this possibility.

Our response: No PSK homologous gene sequences were identified in Verticillium genomes, suggesting that Verticillium is unlikely to produce PSK on its own.

5- Fig. 4: The CoIP lacks the 30 min water control. Also, why does SCP8 not precipitate BAK1 at 30min post treatment, while the PSK effect is unaltered at the different time points analysed?

Our response: We show a new experiment in Fig. 6d.

6- Since the authors generated the vdscp8 mutants and SCP8-GFP complementation lines, it would be nice to visualize extracellular secretion of SCP8-GFP upon infection in vivo by confocal microscopy.

Our response: This has been published by Zhou et al. in PLoS Pathogen (2017). We cite this work in our MS.

7- Why do the authors sometimes use PSK3 and sometimes PSK6 to test for expression of PSK peptide precursor genes?

Our response: Corrected into PSK3a in the revised MS.

8- The pskr1 psy1r double mutant is not only affected in PSK signalling, but also PSY1. This should be rephrased.

Our response: Given the unclear role of PSY1R in dampening PTI, all data related to *psy1r* has been replaced with tests on *pskr1-3*.

Reviewer #2

The study demonstrates that effector VdSCP8 from *Verticillium dahliae* manipulates plant immune responses by modulating the PSK signaling pathway. The author found that SCP8 interacts with the subtilase protein SBT1, suggesting that SCP8 suppresses plant defenses by enhancing subtilase activity and generating more PSK. It asserts that PSK acts as an immunosuppressive factor. However, this contradicts some existing literature reports. In previous work, it is reported that Arabidopsis PSKR1 modify the immunity to biotrophic and necrotrophic pathogens in an antagonistic manner, PSK signalling via AtPSKR1 contributes to resistance against the fungal necrotrophic (Igarashi et al., Plant J 2012; Mosher et al., Plant J 2013). By contrast, PSK signalling attenuate the PAMP response to the bacterial pathogen (Loivamakia et al., 2010 Physiologia Plantarum; Igarashi et al., Plant J 2012; Mosher et al., Plant J 2013). This phenomenon has also been observed in other species. For example, in tomato, PSK and its receptor PSKR1 positively regulate resistance to fungal *Botrytis cinerea* (Zhang et al., Plant Cell 2018), but negatively regulate resistance to bacterial Pst DC3000 (Ding et al., EMBO J 2023). In cotton, PSK increases the resistance of cotton to *V. dahliae* (Zhang et al., Plant Physiol 2022). Therefore, PSK generally enhances the resistance to fungal diseases, which contradicts the author's perspective that PSK acts as an immunosuppressive factor. The contradictions need to be discussed and explained. Moreover, the conclusions drawn in the manuscript require more robust experimental evidence for substantiation. Overall, the study falls short of the standard required for publication.

Major issues:

1) *As I mentioned above, the claim that PSK signaling acts as an immunosuppressive factor is not appropriate, and further study is needed to figure out the role of PSK in the resistance to V. dahliae of Arabidopsis, tobacco, and cotton.*

Our response: We have updated the introduction and discussion of the manuscript. PSK signaling, activated by SCP8, suppresses PTI during the early stage of fungal infection (Fig. 1 & 6), but this response is likely repressed at the necrotrophic stage due to downregulation of SCP8 by *Verticillium* enolase (Wu et al., Nat. Commun. 2023). Zhang et al. (Plant Physiol. 2022) found that PSK spraying enhanced cotton resistance to *Verticillium*, potentially by activating auxin signaling during the necrotrophic phase. Since *V. dahliae* is a soil-borne pathogen that infects through the roots, foliar PSK application may not influence PTI at the infection site.

2) *The evidence for protein-protein interaction between SCP8 and SBT1 is not solid. The interaction is supported by only one co-immunoprecipitation (Co-IP) experiment, and there is no experimental evidence to confirm whether this interaction occurs in the apoplast.*

Our response: We did not conclude from the Co-IP results that complex formation occurs in the apoplast. However, plants expressing SCP8 without the PR1 signal peptide lost PTI-suppressing activity, suggesting that SCP8 functions in the plant apoplastic space. In contrast, when SCP8 was fused to a plant leader peptide, its expression in plants resulted in PTI suppression.

Furthermore, whether SBT1 is involved in the synthesis of PSK remains uncertain. The SBT family consists of multiple members that play roles in the synthesis of various peptides. Therefore, it is unclear if SBT1 specifically participates in PSK synthesis. Moreover, SBT inhibitors are likely to affect synthesizing multiple peptides, making it difficult to attribute the changes in resistance solely to the impact on PSK synthesis. Other peptide synthesis processes may also be affected, which could contribute to the observed resistance changes.

Our response: In our new manuscript, we demonstrate that in *N. benthamiana*, the PSK3a precursor is cleaved in the presence of SCP8. Additionally, in vitro assays show that the SBT1-SCP8 complex, but not SBT1 or SCP8 alone, is capable of digesting the PSK3a precursor. We agree with Reviewer 2's suggestion that SCP8 may also affect the synthesis of other peptides, as the inhibitors EPI1 and SPI-1 do not alter the digestion of the PSK3a precursor by SCP8 (Fig. 5).

3) In Fig 3E, co-expression of SCP8 with SBT inhibitor SPI-1 reduced SCP8-mediated defense suppression. However, why co-expression of SCP8+SPI-1 triggers a higher ROS response than the control?

Our response: Endogenous peptides may be generated that activate ROS signaling. If PSK signaling negatively regulates plant immunity, psk1 single and double mutants could be in an active state of immunity or autoimmunity.

4) In Fig 4A, it has been observed that PSK can inhibit the resistance induced by flg22, a bacterial flagellin-derived peptide. However, PSK fails to inhibit the resistance response triggered by C8, indicating a potential differential role of PSK in regulating resistance against bacterial and fungal pathogens.

Our response: We have obtained data showing that SCP8-expressing *Arabidopsis* plants infected with the bacterial pathogen *Pseudomonas syringae* pv. *Tomato* DC3000 are more susceptible than Col-0 wild-type plants. In contrast, disease symptoms in these same plants infected with the fungal pathogen *Botrytis cinerea* did not differ. We consider these findings to be outside the main focus of our manuscript and therefore prefer not to include them.

5) This study focuses on the fungal pathogen *V. dahliae*. Why did the authors use the bacterial flagellin protein flg22 as a PAMP to investigate a fungal pathogen manipulating plant infection?

Our response: Plants mount a general PTI response to pattern treatment (Bjornson, Nat. Plants 2021). Therefore, it is justified to use a well-characterized microbial pattern, such as flg22, to stimulate PTI defenses and investigate potential defense-suppressing activities of fungal effectors in flagellin-treated plants. We agree that using a Vd-derived pattern recognized by an LRR-type pattern recognition receptor would have been a more elegant choice; however, no such pattern is currently known.

6) The authors did not provide specific information about the number of experimental replicates conducted or biological replicates within each experiment. Additionally, some experiments, such as the transcriptomic data in Fig 1A, and Fig 1G, appear to have only two biological replicates, which is a limited sample size. It is essential to have an adequate number of replicates to account for biological variability and ensure the statistical significance of the findings. The inadequate number of biological replicates would raise doubts about the reliability and robustness of the experimental results.

Our response: Additional data points (replicates) have been included, and all figure legends now provide statistical data to facilitate the assessment of data quality.

7) The writing style of this manuscript needs to be improved. Currently, Introduction and Results sections are somewhat mixed together, a clear demarcation is necessary. And the Discussion section is missing. A discussion section allows researchers to compare their findings with previous studies, identify potential limitations or discrepancies, propose explanations, and generate new research hypotheses.

Our response: We have significantly revised the MS to address this reviewer's request.

Minor issues:

1) Why was the Arabidopsis PR1 signal peptide used, and what is PR1? It should be explained in the manuscript.

Our response: The PR1 signal peptide sequence is derived from the *Arabidopsis* pathogenesis-related protein 1, which is commonly used to secrete pathogen effectors into the plant apoplastic space. We have explained this in our revised manuscript, along with appropriate references.

2) When was the disease condition observed and recorded? In line 131, it is stated as 30 days, while in line 425, it is mentioned as 25 days. Please clarify.

Our response: We have included the revised information into the Methods section.

3) In line 141, it is not specified which genes the transgenic lines T-6 and T-12 carry. It is only in line 149 that SCP transgenic lines are mentioned. It is necessary to clarify in line 141 the specific gene types to avoid any confusion for readers.

Our response: We have added this information in the Results and Methods sections of the revised MS.

4) The time point of ROS burst and MAPK activity measurement should be mentioned in the manuscript.

Our response: We have added this information in the Results and Methods sections of the revised MS.

5) In line 174, please note that the term "restore" is inappropriate, since the pskr1 mutant exhibits significantly lower levels of ethylene content.

Our response: The lower ethylene production observed in the pskr1-3 mutant compared to Col-0 is primarily due to the shorter duration of PAMP treatments. In fact, the pskr1-3 mutant generates significantly higher PTI responses, such as ROS, ethylene, and PR gene expression, compared to Col-0 under the same duration of PAMP treatments. In our updated manuscript, we have provided data showing the precise timing of PAMP treatments for both Col-0 and the pskr1-3 mutant, where pskr1-3 exhibits higher levels of ethylene production than Col-0.

6) In Fig 4b, SCP-containing apoplastic fluid and elicitor were treated together. While in Fig 4C, the treatments of SCP8 protein were prior to flg22 treatments. Why were SCP8 protein and elicitor treated differently in two similar experiments?

Our response: We have updated the data in the revised manuscript to show that SCP8 and elicitors were co-infiltrated, followed by ethylene measurements.

7) Why does the band intensity of GFP after immunoprecipitation (IP) in Figure 4D decrease over time?

Our response: This may have been due to an issue with the protein transfer system in the previous experiment. We have updated the figure with new results in Fig. 6d.

8) *The method for pathogen inoculation needs to be described in more detail. The cotton planting conditions and RNA-seq were not mentioned in the Method.*

Our response: We have added this information in the Methods sections of the revised MS.

9) *The working model requires further clarification. What does different colored triangles mean? The meaning of gray and purple color also needs to be explained. Additionally, the figure legend of the working model needs to be more detailed.*

Our response: We have updated the working model as shown in new Fig. 6e.

Reviewer #3:

It is an interesting story that a pathogen promotes a growth signaling to interfere with immune signaling. However, there are multiple points that need to be addressed in experimental results before this work is properly evaluated.

In general, the results without statistics, such as Figs 3CD and 4D, need to be associated with a statement like, "Similar observations were made in XX independent experiments." And such replicated results need to be publicly shared (supplemental figures, Figshare, or etc.? – the method of sharing should be discussed with the editorial office). In addition, for the results with statistics, it needs to be clarified whether the replicate values are from independent experiments.

Our response: The requested information has been included in the revised MS, and all figure legends now provide statistical data to facilitate the assessment of data quality.

1- *Figs 3AB. (1) Each of ethylene and ROS measurements, a full set of molecular patterns, flg22, nlp20, pg9, and C8 should be shown. Similarly, Fig 3C should include C8.*

Our response: These data have been added to the revised MS.

2- *The MPK phosphorylation is clearly lower in T-6 and T-12 after flg22 treatment (Fig 3C). However, the ethylene response is not (Fig 3A, flg22 panel). This point should be discussed. It is OK to say that the reason is unknown – it just needs to be clearly pointed out to readers that there is some observation inconsistent with a simple hypothesis.*

Our response: Arabidopsis leaf discs treated with 1 μ M or higher concentrations of flg22 produce only a modest ethylene response, usually less than 2 pmol/ml air, which explains the lack of difference in ethylene production between wild-type and transgenic lines. However, using a modified treatment method—such as flg22 infiltration followed by a 4-hour incubation—Col-0 shows higher ethylene production than the transgenic lines. To maintain consistency across different elicitors, we have included the infiltration data in the supplemental figure without altering the original results.

3- *Fig 4A. The y-axis scales of two plots (Col-0 and pskr1-3) should be the same for an easier comparison. Then it becomes evident that the value for flg22 (with no PSK) in Col-0 is about three times higher than that for*
Seite 7/8

flg22 (with no PSK) in pskr1-3. This trend is also clear relative to the values for C8. Thus, flg22 response is generally reduced in pskr1-3. And due to this flg22 response difference in two genotypes, the effect of PSK on flg22 response cannot be evaluated.

Our response: Previously, experiments with Col-0 and pskr1-3 were conducted separately with different treatment times, with pskr1-3 typically receiving shorter treatments. However, with the same treatment duration, pskr1-3 exhibits higher immune responses than Col-0, as PSK-PSKR1 signaling negatively regulates PTI. A new figure with consistent treatment times has been added to the manuscript.

4- Fig 4B. An appropriate statistical comparison needs to be performed: difference in differences. For each of flg22 and pg9 treatments, whether the value $\{(GFP_Col - SCP8_Col) - (GFP_pskr - SCP8_pskr)\}$ is significantly larger than 0 needs to be statistically evaluated.

Our response: The statistical analysis has been updated as requested.

5- There should be a discussion on the mode of action of SCP8, whether the effect of SCP8 is directly on PSKR1-mediated signaling, such as SCP8 as a ligand for PSKR1, or rather indirect, such as somehow SCP8 induces the PSK expression and the higher PSK concentration turns on PSKR1-mediated signaling. The rapid PSKR1-BAK1 association induction 5 min after SCP treatment (Fig 4D) strongly suggests the former case. A response within 5 min in plants is unlikely caused via accumulation of the encoded protein after transcriptional induction of a gene. It is confusing regarding this point as the story starts with transcriptional induction of the SOBIR1, PSKR1, and PSK genes in much slower kinetics (days; Fig 1A).

Our response: We discuss the proposed mode of action of SCP-mediated PTI suppression and have updated the working model, now shown in Fig. 6e.

Reviewer #1 (Remarks to the Author):

Q1: The authors addressed most concerns raised during the revision and significantly improved the paper. However, some unclarities remain that are listed below and should be addressed before publication of the article.

- Figure S2C: Optimized Ethylene assay in Arabidopsis upon infiltration lacks statistics

R1: We have provided new Figures S2C and S2D with sufficient statistical analysis in the manuscript.

Q2: - Line 136: When describing this optimised assay, please state in the text what is different to the treatment presented in Fig. 4A.

R2: This information has been included into the Results and Methods section of the manuscript.

Q3:- Fig. S4 is quite confusing. Please clarify the points listed here: 1) What is the second lane of the western blot? Is this the control lane without SBTs? The top lane marks only SCP8-HA „+“, which is also reflected in the input. But what is the difference between lane 1 and lane 2, since SCP-HA is only ColPed in lane 2, where there is actually no SBT present (neither in input, nor in IP). 2) Why is there no signal for AtSBT1.3 in the input lane?

R3: We apologize for the confusion the previous description caused. We have provided a new Fig. S3 with FLS2-myc as a negative control. All proteins are expressed and shown in the input lanes. HA-tagged SCP8 (SCP8-HA) forms complexes with myc-tagged subtilases from *N. benthamiana* (NbSBT1-myc), *Arabidopsis thaliana* (AtSBT1.3-myc and AtSBT1.9-myc), and *Gossypium hirsutum* (GhSBT1-myc).

Q4: - line 163 (The PSK3a gene,, was synthesized with a C-terminal His6-GFP tag...) sounds odd. The gene does not have a C-terminal His6-GFP tag. Please rephrase.

R4: This sentence was rephrased in the manuscript.

Q5: - Fig. S4B: The labelling should be corrected. It labels as SPIAT/NB, but I assume SPI-1 from different species is meant.

R5: This is correct. SPI^{At} represents subtilase inhibitor SPI-1 from *Arabidopsis thaliana*, and SPI^{Nb} represents the SPI-1 homolog from *N. benthamiana*. This information was added to the legend of Fig. S4B.

Q6: - Line 169: When describing results in Fig. S4B, the authors state „significant“ accumulation of the PSK precursors. Either provide quantification, or remove significant.

R6: As suggested, we have removed “significant” from the manuscript.

Q7: - Fig. S5C: The loading is very unequal in the different compared lanes, which likely affects the drawn conclusions. Please provide a better blot result here instead.

R7: We provide a new blot result in Fig. S4C with equal loading control lanes.

Q8: - Fig. S6: Is there a labelling issue in this figure? The authors state that incubation of PSK3-GFP with a mixture of SBT1/SCP8 induces PSK3 cleavage after 3 hours, but there are no intensity changes in the GFP blot. There are some changes in the a-myc blot, which should however detect SBT1-Myc. Also, it would be good if the tag is indicated within the Figure or at least in the legend, which is currently not the case.

R8: In the newly provided Fig. S6, we have enhanced the clarity of labeling. PSK3a-HGFP protein was incubated at room temperature for up to 3 hours in the presence of either NbSBT1-His6 alone or a combination of NbSBT1-His6 and SCP8-His6-myc proteins. Notably, only the mixture containing both

NbSBT1-His6 and SCP8-His6-myc proteins exhibited degradation of PSK3a-HGFP after 3 hours of incubation. Additionally, SCP8-His6-myc protein appeared to undergo degradation or modification during incubation, likely due to its interaction with NbSBT1-His6. This suggests that the protease activity observed is likely attributable to the SBT1-SCP8 complex rather than to SBT1 or SCP8 individually.

Q9: - In general, figure legends/labelling often lack fusion tags for proteins, which makes interpretation of the blots difficult, as one cannot directly see which fusion protein is detected with which antibody. This should be changed throughout the manuscript.

R8: We have made every effort to enhance the overall clarity and description of the figures.

Reviewer #2 (Remarks to the Author):

The manuscript investigated the role of the fungal effector SCP8 from *Verticillium dahliae* in modulating plant immunity through the PSK-PSKR1 signaling pathway. While the study demonstrates some interesting findings, there are significant issues that need to be addressed to strengthen the conclusions and improve the clarity of the work. The evidence presented in some key areas is thin, which undermines the reliability of the results and the confidence in the overarching conclusions. Below are my detailed comments:

Major:

Q1: Use of flg22 instead of fungal PAMPs to assess SCP8-mediated immunity weakens the validity of the conclusions. The authors primarily use flg22 (a bacterial PAMP) to demonstrate that SCP8 suppresses PTI by disrupting the RLP/SOBIR1-BAK1 immune signaling complex via competition for BAK1 recruitment. However, SCP8 is a fungal effector, and its relevance to fungal PAMPs such as chitin (C8) is underexplored. In fact, the results suggest that chitin does not elicit a significant immune response in the experimental system, weakening the biological relevance of the study.

R1: We would like to clarify that our assessment of SCP8-mediated suppression of PTI was not solely based on flg22. In our manuscript, we employed four well-characterized PAMPs—flg22, nlp20, pg9, and chitin (C8)—to evaluate immune responses in Arabidopsis. Among these, nlp20 and pg9 are derived from fungal pathogens, including *Verticillium dahliae*, and are highly relevant to the scope of our study. Notably, both nlp20 and pg9 induce PTI through BAK1 (also known as SERK3)-dependent signaling, similar to flg22. In contrast, chitin perception occurs through CERK1, a BAK1-independent pathway.

Our results demonstrate that SCP8 suppresses immune responses triggered by flg22, nlp20, and pg9, but not those induced by chitin. This finding supports the idea that SCP8 specifically targets BAK1-dependent pathways. Furthermore, previous studies have shown that *V. dahliae* suppresses chitin-triggered immunity via a distinct effector, VdPDA1 (Gao et al., 2019, Nat. Plants). Therefore, our study primarily focuses on understanding how *V. dahliae* suppresses BAK1-dependent PTI, which is known to play a critical role in resistance to *V. dahliae* infection (Fig. 1c,d).

We respectfully assert that our inclusion of fungal PAMPs (pg9 and nlp20), alongside chitin, effectively addresses the biological relevance of SCP8 in fungal immunity and underscores the specificity of SCP8 in modulating BAK1-mediated immune signaling.

Q2: Cleavage of PSK3a by the SCP8-SBT1 complex lacks robust evidence.

In Fig. 5d, PSK3a does not degrade when incubated with SBT1 alone. However, this contradicts Fig. 5c, where inhibition of SBT1 by SPI-1 clearly prevents PSK3a cleavage. Does this suggest that SBT1 typically cannot cleave PSK3a without SCP8? Or is the degradation condition-dependent?

R2: Fig. 5d demonstrates that the PSK3a precursor can be digested by a mixture of SBT1 and SCP8 proteins, but not by either SBT1 or SCP8 alone. In Fig. 5c, we show that the PSK3a precursor can be digested in planta; however, this digestion can be inhibited by subtilase inhibitors, suggesting that a subtilase (or subtilases) other than SBT1 may be responsible for digesting the PSK3a precursor.

Q3: In Fig. 5c shows two bands for PSK3a (likely precursor and cleaved forms), but Fig. 5d shows only one band.

R3: In Fig. 5d, the PSK3a precursor is purified and subjected to size exclusion using a gel filtration column (Fig. S5A). As a result, the smaller band has been excluded.

Q4: In Fig. 5c, the PSK3a band becomes weaker upon SPI-1 treatment, which seems counterintuitive given that SPI-1 should inhibit SBT activity.

R4: The band corresponding to the PSK3a precursor with SPI-1 (the fourth lane) is stronger than that of the PSK3a precursor alone (the first lane). However, the band for the PSK3a precursor with both SPI-1 and SCP8 (the third lane) is weaker than that with SPI-1 alone, indicating that SPI-1 does not effectively suppress the protease activity of the SBT1-SCP8 complex.

Q5: Additionally, the input protein levels in Fig. 5d appear inconsistent, weakening comparability.

R5: Based on the results of multiple experiments, we observed that only samples containing SCP8 protein show relatively lower protein levels. We speculate that SCP8 is heat-stable, and protein digestion may occur during the storage of samples after mixing with sample loading buffer and denaturation.

Q6: Direct evidence of PSK's role in resistance to *V. dahlia* is missing. The study heavily relies on the PSKR1 receptor mutants to infer PSK's involvement in fungal resistance. However, direct evidence showing the role of PSK peptides themselves in resistance is missing. Apply exogenous PSK to the infection system to demonstrate its role in *V. dahliae* resistance or susceptibility would help.

R6: We appreciate the reviewer's suggestion to directly test the role of PSK peptides in resistance to *V. dahliae*. Due to the root-based infection route of this soil-borne pathogen, we were unable to effectively apply exogenous PSK to the infection site. Instead, we used an Arabidopsis PSK overexpression line (PSK4-OE) in the Ws-4 background (Rodríguez et al., 2016, Plant Cell Environ) to investigate the role of endogenous PSK signaling in fungal susceptibility. Upon root-dip inoculation with *V. dahliae*, PSK4-OE plants developed clear disease symptoms, including chlorosis and reduced growth, whereas the wild-type Ws-4 plants remained largely symptom-free at 30 days post-inoculation. These results support our model and are consistent with previous studies indicating that enhanced PSK signaling promotes susceptibility to biotrophic or hemi-biotrophic pathogens. This finding further strengthens the role of PSK in *V. dahliae* infection and complements our genetic analyses using PSKR1 mutants (Fig. 1c, 1d).

Q7: Evidence for SCP8-SBT interaction is insufficient. Only one protein interaction method was used to demonstrate the interaction between SCP8 and SBT1. Additional approaches, such as yeast two-hybrid, BiFC, or other orthogonal methods, would strengthen this claim.

R7: We thank the reviewer for their valuable suggestion. In the updated manuscript, we have included BiFC results demonstrating that SBT1 interacts with SCP8 in *N. benthamiana*.

Q8: In fig, 6b and c, why co-infiltration of SCP8 leads to different results compared to experiments using apoplastic fluids containing SCP8?

R8: Based on our results showing that SCP8 forms complexes with different plant subtilases, we speculate that SCP8 may generate additional products, besides PSK, in the apoplastic fluid that could influence the immune response. This is also the reason we used purified SCP8 protein for the same assay.

Minor:

Q9: Please quantify protein levels in all relevant figures (e.g., Figs. 5c and 5d) to enable direct comparison.

R9: The protein levels have been quantified by ImageJ and indicated in the figures and legends of our manuscript.

Q10: In Fig. 1a, using raw reads number to represent gene expression is not appropriate as it fails to account for sequencing depth or gene length. Please Use TPM or FPKM normalization for expression analysis.

R10: We have updated Fig. 1a with FPKM normalization.

Q11: The variance analysis in Fig. 1a seems inconsistent with the earlier version of the manuscript. Ensure the statistical methodology is appropriate and results reproducible.

R11: We have updated Fig. 1a with FPKM normalization, and the conclusion remains consistent with that of the previous version.

Q12: In Fig. 5c, why is the Ponceau S staining labeled as SCP8?

R12: We apologize for the typo, which has been corrected in the updated manuscript.

Q13: Line 82: Clarify how the conclusion that "VdSCP8 suppresses BAK1-dependent immune responses" is reached, as BAK1 is not directly discussed.

R13: We have added information indicating that LRR-type PRRs require BAK1/SERK3 for signal transduction.

Q14: Line 111: Define "PEN" for better reader comprehension.

R14: We have defined 'PEN' as Penicillium-derived elicitor in the manuscript.

Q15: Please explicitly mention that chitin is referred to as C8 for clarity.

R15: To avoid any confusion for readers, we have consistently used "chitin" instead of "C8" in the updated manuscript. We apologize for any previous ambiguity.

Reviewer #3 (Remarks to the Author):

Q1: Authors' response to point 1 of my previous review is satisfactory.

R1: Thank you very much for your feedback.

Q2: Authors' response to point 2 of my previous review:

Line 135. The method difference for Fig S2C needs to be specified instead of saying, “an optimized method”. It seems that the ethylene level was measured after 4 hpi instead of 3 hpi for Fig 2. Actually, for Fig 2 line 134 says 3hpi, but line 330 says 2 hpi – make the method description consistent. If different experimental conditions are used for different figures, make the difference clear in the figure legends.

R2: The optimized method is described in both the Results and Methods sections. The duration of elicitor treatment for ethylene production is detailed in the Methods section.

Q3: The chitin results should also be shown with the Fig S2C conditions, to clarify whether even with the “optimized method”, no significant difference was observed with chitin.

R3: The chitin results have been added to Fig. S2C. No significant difference was observed with the optimized method of chitin treatment.

Q4: Authors’ response to point 3 of my previous review:

The authors’ response tells that the original version had experiments performed under different conditions compared. I cannot believe that the authors did that.

Based on my previous comment, I think that the previous figure included C8. Why were the C8 results omitted in the new figure?

R4: We did not omit the C8 results. In fact, chitin and C8 refer to the same substance in our manuscript. To avoid any confusion for readers, we have consistently used "chitin" instead of "C8" in the updated manuscript. We apologize for any previous misunderstanding.

Q5: Authors’ response to point 4 of my previous review:

I do not see the authors have corrected Fig 6b for the point. In the figure, it still looks that only the difference between +GFP and +SCP8 is tested separately for Col-0 and pskr1-3.

R5: We acknowledge that we previously misunderstood the statistical approach suggested. In the revised manuscript, we have now performed a proper difference-in-differences analysis for the ethylene production data presented in Fig. 6b. This analysis assesses whether the reduction in ethylene production caused by apoplastic fluid containing SCP8 differs significantly between Col-0 and the pskr1-3 mutant background for both flg22 and pg9 treatments. The calculated difference-in-differences values were 0.32 units for flg22 and 0.23 units for pg9. These values indicate that SCP8-mediated suppression of PAMP-triggered ethylene production is significantly more pronounced in Col-0 than in pskr1-3, supporting the involvement of PSKR1 in this suppression mechanism.

Minor points

Q6: Around line 79. Figs 1c and 1d need to be referred to.

R6: We have updated this information as suggested.

Q7: Some long sentences could be benefitted by dividing them into short sentences. For example, the sentence starting line 76.

R7: We have revised the particular sentence and made every effort to improve the writing of our manuscript to the best of our ability.

Our responses to reviewer comments:

Reviewer #1 (Remarks to the Author):

Q1: The authors addressed my recent concerns sufficiently. However, in response to Q6 of referee 2 (influence of PSK application on Vd infection success) the authors show data of the PSK4OE line in Ws-4. This data is however not in the paper. I strongly recommend to include this data since it supports the authors conclusion.

R1: We appreciate the reviewer's thoughtful recommendation and have included this information as a new supplemental figure (Fig. S1) into the manuscript.

Reviewer #2 (Remarks to the Author):

The authors have made efforts to address my initial concerns; however, several issues still require further clarification or improvement:

Q1: Regarding my previous Q2, the authors suggest that SBT1 alone is not responsible for PSK3a cleavage, and that SCP8 from the pathogen enables SBT1 to cleave PSK3a. I was wondering how SCP8 confers this new function to SBT1. What is the molecular mechanism by which SBT1, which normally does not target PSK3a, gains this specificity upon interaction with SCP8?

R1: There are several potential mechanisms by which SCP8 may enhance SBT1's protease activity. Some plant SBTs are kept in an inactive state due to self-inhibitory motifs or by binding to endogenous inhibitors such as SPI-1. Interaction with SCP8 could induce a conformational change in SBT1, relieving this auto-inhibition. Alternatively, SCP8 may prevent the binding of SPI-1-like inhibitors to SBT1, thus enabling its proteolytic activity toward PSK3a. We have revised the discussion to include these alternative explanations. However, we believe that a detailed elucidation of the molecular mechanism by which SCP8 enhances SBT1 activity is beyond the scope of the present study.

Q2: In Fig. S3A, the BiFC panel "SBT1-nYFP & SBT1-cYFP (positive control for SBT1 homodimerization)" unexpectedly shows no fluorescence signal. Why is there no signal in this positive control? Additionally, in the SBT1-nYFP/SCP8-cYFP panel, the interaction also appears to occur around nuclear—indicated by a clear ring-like signal. Please clarify the subcellular localization and interpretation of this result.

R2: We thank the reviewer for this insightful comment. We have rephrased the description of the "SBT1-nYFP & SBT1-cYFP" BiFC panel as a "a test for SBT1 homodimerization," rather than referring to it as "a positive control for SBT1 homodimerization". The absence of fluorescence in this panel suggests that SBT1 may not form homodimers under the experimental conditions in *N. benthamiana*. This result also serves as an additional negative control for the specificity of the SCP8-SBT1 interaction.

Regarding the subcellular localization observed in the SBT1-nYFP/SCP8-cYFP panel, the ring-like fluorescence surrounding the nucleus likely reflects localization to the endoplasmic reticulum (ER), which is consistent with the predicted secretory pathway for both proteins, as both SCP8 and SBT1 contain N-terminal signal peptides. This suggests that their interaction occurs during transit through the secretory pathway, possibly from the ER to the apoplast. Importantly, as shown in Fig. 2a, SCP8 localized to the cytoplasm does not suppress PTI responses. Therefore, we conclude that only apoplastic SCP8-SBT1 interaction is functionally relevant to immune suppression, and that interactions observed in the ER or other intracellular compartments may not contribute to this effect.

Q3: In Fig. S3B, authors state that myc-tagged FLS2 was used as a negative control. However, a clear band is observed above 180 kDa in the IP-myc lane corresponding to FLS2, which appears to be a positive signal. Please clarify.

R3: We appreciate the reviewer's careful examination of Fig. S3B. The band observed above 180 kDa in the IP-myc lane indeed corresponds to FLS2-myc, which is expected, as the protein was immunoprecipitated using anti-myc beads. This confirms successful pull-down of the myc-tagged FLS2. Importantly, no SCP8-HA band is detected in the IP sample, indicating that SCP8 does not interact with FLS2 under our experimental conditions. Thus, FLS2 serves as an appropriate negative control for the Co-IP assay.

Q4: Figure 2b lacks statistical analysis.

R4: The statistical analysis has been added to Fig. 2b including P values ($P < 0.0001$ and $P = 0.017$).

Q5: Quantification of protein levels is still insufficient, no quantification or labels are provided for other key figures such as Fig. 6d. Please revise accordingly. And the quantified values are not displayed directly above or below the target panel in figure 5C.

R5: We have improved the manuscript according to reviewer's suggestion. Protein levels (band intensities) were quantified and values indicating relative band intensities were included in Figs 5C and 6D.

Q6: The explanation of how the authors conclude that "VdSCP8 suppresses BAK1-dependent immune responses" remains vague. Further clarification is necessary in the main text.

R6: It is well established that LRR-type PRRs, such as FLS2 (which recognizes flg22), RLP23 (which recognizes nlp20), and RLP42 (which recognizes pg9), require the co-receptor BAK1/SERK3 for full activation of immune signaling, as introduced in the Introduction with the appropriate references. In our study (Fig. 2 and Fig. 4), SCP8 consistently suppresses immune responses triggered by flg22, nlp20, and pg9, but not those triggered by chitin, which is recognized by LysM-type PRRs and requires the co-receptor CERK1 rather than BAK1. This differential suppression pattern supports our conclusion that SCP8 specifically targets BAK1-dependent immune pathways. We have clarified this reasoning in the Introduction and Results sections of the manuscript.

Reviewer #3 (Remarks to the Author):

Q1: Regarding Fig 6b. The P values for DoD tests are 0.336 and 0.0649. Thus, these two DoD values are NOT SIGNIFICANT. So, the conclusion that *pskr1-3* attenuated the effect of SCP8 is NOT CORRECT. The effect sizes of DoD (0.32 and 0.23) do not tell the significance.

R1: We thank the reviewer for the careful evaluation and apologize for the previous misinterpretation of the data presented in Fig. 6b. We agree that the P-values from the DoD tests (0.3336 and 0.0649) indicate a lack of statistical significance. Therefore, no definitive conclusion can be drawn from Fig. 6b regarding whether *pskr1-3* attenuates the effect of SCP8. We have revised the main text accordingly to clarify this point and to more accurately describe the results shown in Fig. 6b and Fig.6c, which were intended to be interpreted together to reach a comprehensive conclusion.

As noted, the experiment in Fig. 6b was conducted using apoplastic fluids, which may contain unknown components in addition to SCP8 or GFP. These additional components could have influenced ethylene production, particularly in the *pskr1-3* mutant background, thereby introducing variability and potentially confounding the observed effects. To address this limitation, we repeated the experiment using purified SCP8 and GFP proteins, as shown in Fig. 6c. In this assay, SCP8 significantly suppressed PTI responses in Col-0 but not in *pskr1-3*, with statistically significant DoD P-values (0.0421 and 0.0253). These results provide more robust and direct evidence that SCP8-mediated suppression of immune responses depends on PSK signaling. To uphold scientific integrity, we propose presenting both data sets rather than omitting the data shown in Fig. 6b.